# Bandit Social Learning with Exploration Episodes

**Kiarash Banihashem** [1]  **Natalie Collina** [2]  **Aleksandrs Slivkins** [3]

## Abstract

We study a stylized social learning dynamics where self-interested agents collectively follow a simple multi-armed bandit protocol. Each agent controls an "episode": a short sequence of consecutive decisions. Motivating applications include users repeatedly interacting with an AI, or repeatedly shopping at a marketplace. While agents are incentivized to explore within their respective episodes, we show that the aggregate exploration fails: e.g., its Bayesian regret grows linearly over time. In fact, such failure is a (very) typical case, not just a worst-case scenario. This conclusion persists even if an agent's per-episode utility is some fixed function of the per-round outcomes: e.g., `min` or `max`, not just the sum. Thus, externally driven exploration is needed even when some amount of exploration happens organically.

## 1. Introduction

We consider a common type of social learning dynamics with a feedback loop. Self-interested users/customers (henceforth, *agents*) choose among available alternatives, under some uncertainty on the quality of these alternatives, and return feedback on their experiences. This feedback is aggregated by an online platform and served to the future agents, affecting their decisions. Collectively, the agents face *exploration-exploitation tradeoff*,[1] but individual agents favor exploitation, reluctant to explore for the sake of others.

This misalignment of incentives often leads to aggregate

---

K.B. and N.C. were interns at Microsoft Research during the active phase of this research project.

[1]University of Maryland, College Park, MD, USA. [2]University of Pennsylvania, Philadephia, PA, USA. [3]Microsoft Research, New York, NY, USA. Correspondence to: Kiarash Banihashem <kiarash@umd.edu>, Aleksandrs Slivkins <slivkins@microsoft.com>.

*Proceedings of the $43^{rd}$ International Conference on Machine Learning*, Seoul, South Korea. PMLR 306, 2026. Copyright 2026 by the author(s).

[1]This tradeoff – between experimentation and making optimal near-term decisions – is a central and extremely well-studied issue in multi-armed bandits and reinforcement learning.

inefficiency. In fact, agents may converge on suboptimal alternative(s), essentially giving up on further exploration. Such outcomes, called *learning failures*, are widely observed empirically and are very typical in theory. In fact, they provably happen (with positive probability) as a very common case in a wide variety of underlying learning problems (Banihashem et al., 2023; Slivkins et al., 2025).

While prior work mostly focused on agents making one decision each, we allow each agent to make *several* decisions (as is common in practice). The key difference is that agents may now explore for the sake of improving their own future decisions. This exploration adds up over time, even if each agent explores very little, possibly side-stepping learning failures mentioned above. Thus, we ask:

(Q) Can learning failures be avoided when self-interested agents make a few decisions each?

Despite the failure results mentioned above, we emphasize that learning failure is *not* a foregone conclusion for exploitation-only learning dynamics. Indeed, such dynamics may converge to the best alternatives (and, under some conditions, even learn optimally fast) if the underlying learning problem exhibits a suitable structure (*e.g.,* Kannan et al., 2018; Bastani et al., 2021; Slivkins et al., 2025).

For a concrete example tailored to (Q), suppose there are two arms with (unknown) deterministic payoffs, each agent makes two consecutive decisions, and her aggregate utility from these decisions is the largest of the two rewards. Then the first agent tries both arms (receiving the maximal possible utility), after which both arms are known.

**Our contributions.** We investigate (Q) in a stylized multi-armed bandit scenario. There are two arms, each associated with a fixed but unknown reward distribution with bounded rewards. In each round, an arm is chosen and its random reward is realized and observed.[2] Time is partitioned into *episodes* of $m \geq 2$ bandit rounds each. In each episode, a new agent arrives and controls decisions within this episode. To avoid degenerate outcomes when agents explore despite deriving little or no utility from doing so, we posit *selection*

---

[2]When each round is controlled by an algorithm, this is the quintessential case of multi-armed bandits, as they are studied in machine learning and computer science.

*costs* (which are typically very small), and an option to skip rounds (getting zero payoff). The overall framing is Bayesian: there is a common Bayesian prior, and each agent chooses a policy (*i.e.,* a bandit algorithm for her episode) that is Bayesian-optimal given the current data.

The case of single-round episodes ($m = 1$) corresponds to "pure exploitation" in Bayesian bandits and leads to learning failures (Banihashem et al., 2023), whereas any $m \geq 2$ allows for within-episode exploration.

Somewhat surprisingly, we find that learning failures are the typical case, for any fixed episode length $m$. Essentially, learning failures happen with positive probability *for every problem instance*, not just as a worst case. It follows that Bayesian regret of the learning dynamics scales linearly as a function of time. In contrast, sublinear Bayesian regret is a standard "minimal desiderata" for a useful bandit algorithm.

This conclusion is robust to the choice of agents' utility model. An agent's aggregate utility for the episode is, in general, some fixed function of rewards received during the episode. The most natural options are the sum ("all rewards matter"), the $\max$ ("even one good round suffices"), and the $\min$ ("can't tolerate a mistake"). We handle these three options, and in fact allow an arbitrary function $f$ of the rewards under mild conditions. Namely, $f$ must be non-constant, non-decreasing in each coordinate, and symmetric. We also provide a similar result for non-symmetric $f$, restricted to episode length $m = 2$.

We also provide partial extensions that go beyond two arms and touch upon the case of no selection costs (Section 5).

**Additional motivation.** We zoom in on two features of bandit-like social learning dynamics: agents making multiple decisions and evaluating the joint outcome of these decisions in different possible ways. While our model is extremely stylized, these two features are present in several application scenarios.

First, consider human users repeatedly interacting with an AI, *e.g.,* via prompting, and sharing their interaction strategies and experiences with others via the AI platform or social media. A given user may make multiple attempts to accomplish a specific task, *e.g.,* write a CV or generate an image, and choose the best one. This corresponds to $f = \max$. Alternatively, a user may have a workload consisting of many instances of the same task type, *e.g.,* use AI for coding; or process many job applications. This maps to fairly generic symmetric functions $f$. [3]

Second, customers repeatedly shopping for products or ex-

periences. This may motivate a variety of functions $f$, depending on complementarities between the products.

Third, real-world activities such as autopilot-assisted driving (which conceptually decomposes into many small instances of driving). This corresponds to $f = \min$ if one does not wish to crash, or $f = \text{sum}$ if one values the average driving experience, or anything in between.

In all these scenarios, users/customers may explore within relatively short bursts of activity, corresponding to the episodes in our model, but may be mostly myopic over longer time scales. So, a new burst of activity of the same agent can reasonably be interpreted as a new agent.

### 1.1. Related Work

Multi-armed bandits is a vast subject, background can be found in (Slivkins, 2019; Lattimore & Szepesvári, 2020). Most relevant to this paper are bandit problems with fixed but unknown reward distributions (a.k.a. *stochastic bandits*), studied since Lai & Robbins (1985); Auer et al. (2002a). We consider a basic version without any "reward structure" that relates arms to one another. A large literature on stochastic bandits under various reward structures is outside our scope. Stochastic bandits with a Bayesian prior (a.k.a. *Bayesian bandits*) is also a fairly standard model, often/usually studied in conjunction with a specific approach called *posterior sampling*, a.k.a. *Thompson Sampling* (Russo et al., 2018).

The social learning dynamics described above can be interpreted as a "greedy" bandit algorithm (Greedy) which only exploits. As such, it has been believed to perform poorly, so that exploration is needed. [4] A classic example to substantiate this belief considers a two-armed bandit problem with 0-1 rewards, and posits that the greedy algorithm is initialized with a single sample of each arm. If the best arm initially returns a $0$ while another arm returns a $1$, Greedy gets permanently stuck on the bad arm. Thus, we have a type of a learning failure discussed above.

Banihashem et al. (2023) extend this example in various directions, proving that similar learning failures (i) hold with substantial probability even if the greedy algorithm is initialized with *many* samples; (ii) happen even if the agents exhibit modest amounts of optimism/pessimism and/or other behavioral biases; and (iii) extend to Bayesian bandits. Slivkins et al. (2025) investigated Greedy for bandit problems with an arbitrary known structure and characterized whether learning failures happen. One take-away message is that positive-probability learning failure is the typical case for $K$-armed bandits. Earlier work considered Greedy for bandit problems with specific one-dimensional reward structures, and derived learning failures for these problems

---

[3] In particular, $f = \text{sum}$, *e.g.,* deriving utility from each AI-assisted piece of code; $f = \min$, *e.g.,* all applications must be processed fairly; or $f = \max$, if spotting one great candidate suffices; or any combination of the above.

[4] Accordingly, most of literature on multi-armed bandits is dedicated to balancing exploration and exploitation.

(Harrison et al., 2012; den Boer & Zwart, 2014; Lai & Robbins, 1982; Keskin & Zeevi, 2018).

Most notable *positive* results for `Greedy` focus on linear contextual bandits (*i.e.,* bandit problems in which a *context vector* is revealed before each round, and rewards are linear in this vector). `Greedy` achieves near-optimal regret rates if the context vectors are sufficiently diverse or smoothed (Kannan et al., 2018; Bastani et al., 2021; Raghavan et al., 2023). In a different direction, `Greedy` is also known to attain $o(T)$ regret in various scenarios with a very large number of near-optimal arms (Bayati et al., 2020; Jedor et al., 2021). Slivkins et al. (2025) derives "asymptotic success" (*i.e.,* sublinear regret) for various reward structures in contextual bandits, and in fact characterizes asymptotic failure vs success for arbitrary finite structures.

An explicit connection between exploitation in a bandit-like environment and social-learning dynamics of self-interested agents is made in Kremer et al. (2014); Che & Hörner (2018) and the subsequent the literature on *incentivized exploration* (Slivkins, 2023). The perspective is different, though: instead of studying a *given* learning dynamics induced by self-interested behavior, they design interventions to change the dynamics and incentivize the agents to explore.

The line of work on "strategic experimentation", starting from Bolton & Harris (1999); Keller et al. (2005), see Hörner & Skrzypacz (2017) for a survey, studies long-lived agents facing the same bandit problem and observing actions and rewards of one another. While each agent can make multiple decisions, like in our model, their scenario is quite different: the agents live forever and act in parallel (arriving at random times). In particular, learning failures cannot happen, essentially because each agent alone can eventually learn the best arm. The technicalities are different, too: one of the two arms is known, and rewards are time-discounted. The agents engage in a complex repeated game where they explore but prefer to free-ride on others. The focus is on how much exploration happens in the equilibria of this game.

## 2. Model and Preliminaries

Self-interested agents collectively follow a simple bandit protocol. There is a fixed set of $K$ arms. Each arm $i$ is associated with an (unknown) reward distribution $D_i$. In each round $t \in \mathbb{N}$, an arm $i = i_t$ is chosen by an agent (as defined below) and the respective reward $r_t \sim D_i$ is realized and observed. Before making the decision, the agent observes the history for all previous rounds, $H_t := \big((i_s, r_s) : s \in [t-1]\big)$, called the *round-$t$ history*. Agents do not receive any other information, *e.g.,* no private signals. If all agents were controlled by an algorithm, this would be a standard $K$-armed bandit problem with stochastic rewards.

Each agent controls $m$ consecutive rounds. Formally, time

is partitioned into *episodes*: sequences of $m$ consecutive rounds. In each episode $e \in \mathbb{N}$, a new agent arrives and controls the decisions within this episode. Equivalently, agent $e$ chooses a *per-episode policy* $\pi_e$: a bandit algorithm for making decisions throughout the episode, possibly adapting to the observations. Before choosing $\pi_e$, the agent observes the current history, *i.e.,* the history of all past agents. The case of single-round episodes ($m = 1$) corresponds to the "greedy" bandit algorithm, as discussed in Related Work. This paper focuses on episode length $m \geq 2$.

We call this problem *Episodic Bandit Social Learning* (`EpiBSL`).[5] Below we spell out some details on rewards and utilities, which are included into `EpiBSL` by default.

**Arms and rewards.** There are two arms $i \in \{1, 2\}$, each with a Bernoulli reward distribution with (unknown) mean $\mu_i \in [0, 1]$. Initially, each $\mu_i$ is drawn independently from a known Beta prior $\mathcal{P}_i = \text{Beta}(\alpha_i, \beta_i)$, where $\alpha_i, \beta_i > 0$. There's also a third arm, called the *skip-arm*, which is known to always yield reward 0 and corresponds to skipping the round. The other two arms will be called *non-skip arms*.

**Agent utility.** Rewards received by a given agent $e$ throughout the episode are aggregated as follows. Let

$$\vec{r}_e := \big(r_{t+(e-1)m} : t \in [m]\big) \in \{0, 1\}^m$$

denote the reward vector for this episode. The aggregate reward (a.k.a. *score)* of agent $e$ is defined as $f(\vec{r}_e)$, for some known *aggregation function* $f \colon \{0, 1\}^m \to [0, m]$. The paradigmatic examples are the sum, the $\max$ and the $\min$ over the observed rewards.

We consider a generic $f$, subject to three mild properties:

- $f$ is coordinatewise non-decreasing,
- $f$ is non-constant (*i.e.,* $f(\vec{0}) < f(\vec{1})$),
- $f(\vec{0}) = 0$ (it's w.l.o.g.: replace $f(\cdot)$ with $f(\cdot) - f(\vec{0})$).

We assume these properties without further mention.

Our main result posits that $f$ is symmetric, *i.e.,* permuting the coordinates in $\vec{r} \in \{0, 1\}^m$ does not change $f(\vec{r})$. [6]

Agents also incur selection costs (henceforth, simply *costs*), defined as some known $c_{\text{sel}} \in (0, 1]$ each time a non-skip arm is selected, and 0 for the skip-arm. Finally, agent's utility $U_e$ is defined as the score $f(\vec{r}_e)$ minus the total cost.

**Agent behavior.** Each agent $e$ chooses its policy $\pi_e$ so as to maximize its Bayesian-expected utility given history. Formally, let $U(\pi_e)$ be the agent's utility under policy $\pi_e$. Then

---

[5]Banihashem et al. (2023) studied the case $m = 1$ from the social learning perspective, calling it "Bandit Social Leaning".

[6]We also provide a result, Theorem 4.3, when $f$ need not be symmetric. This result is limited to the case $m = 2$.

policy $\pi_e$ is chosen as a per-episode policy that maximizes $\mathbb{E}\left[U(\pi_e) \mid \widetilde{H}_e\right]$, where $\widetilde{H}_e$ be the history at the start of the episode. We assume w.l.o.g. that policy $\pi_e$ is deterministic.

We emphasize that each agent acts in its own interest, ignoring the interests of the other agents. In particular, the agents would not cooperate to run any bandit algorithm that deviates from the self-interested behavior proscribed above.

**Problem instance.** Thus, an instance of `EpiBSL` is specified by episode length $m$, joint prior $\mathcal{P} = (\mathcal{P}_1, \mathcal{P}_2)$, aggregation function $f$, and selection cost $c_{\text{sel}}$. The "realized" problem instance contains all of the above and the vector of realized mean rewards, $\vec{\mu} = (\mu_1, \mu_2)$.

### 2.1. Regret

Agents' collective behavior on a given problem instance comprises a bandit algorithm, and therefore can be evaluated as such. We use standard notions of regret from bandit literature, suitably instantiated to `EpiBSL`. Let $V(\pi \mid \vec{\mu}) = \mathbb{E}[U(\pi) \mid \vec{\mu}]$ be the expected utility of per-episode policy $\pi$ given $\vec{\mu}$, a specific realization of the mean rewards. Let $U^*(\vec{\mu}) = \sup_\pi V(\pi \mid \vec{\mu})$ the supremum of that over all policies $\pi$; this is our comparator. The *pseudoregret* after $T$ episodes is defined as

$$\text{Reg}(T \mid \vec{\mu}) := T \cdot U^*(\vec{\mu}) - \sum_{e \in [T]} V(\pi_e \mid \vec{\mu}). \quad (1)$$

We are interested in *Bayesian regret*,

$$\text{BReg}(T) := \mathbb{E}[\text{Reg}(T \mid \vec{\mu})].$$

**Background.** Eq. (1) specializes to the standard notion of pseudo-regret in stochastic bandits when $m = 1$. Optimal regret rates $\text{Reg}(T \mid \vec{\mu})$ for bandit algorithms scale as $O\left(\frac{\log T}{|\mu_1 - \mu_2|}\right)$ for a particular problem instance (Lai & Robbins, 1985; Auer et al., 2002a), and as $\widetilde{O}\left(\sqrt{T}\right)$ in the worst case over all problem instances (Auer et al., 2002a;b).

### 2.2. Preliminaries

We emphasize that a newly arrived agent has exactly the same information as the previous agent at this time. In particular, their Bayesian posteriors coincide.

We leverage well-known facts about conjugate Beta priors. First, Bayesian posteriors are also Beta distributions. Fix a non-skip arm $i$ and round $t$. Let $S_{i,t}$ be the total reward from this arm after round $t$. Then Bayesian posterior of $\mu_i$ after round $t$ is $\mathcal{P}_{i,t} = \text{Beta}(\alpha_{i,t}, \beta_{i,t})$, where $\alpha_{i,t} = \alpha_i + S_{i,t}$ and $\beta_{i,t} = \beta_i + (t - 1 - S_{i,t})$. Second, the prior-mean reward for arm $i$ is $\mathbb{E}[\mu_i] = \frac{\alpha_i}{\alpha_i + \beta_i}$. Likewise, the posterior-mean reward of arm $i$ after round $t$ is $\mathbb{E}[\mu_i \mid H_t] = \frac{\alpha_{i,t}}{\alpha_{i,t} + \beta_{i,t}}$.

Throughout, we posit w.l.o.g. that arm 1 is *prior-good*, *i.e.,* $\mathbb{E}[\mu_1] > \mathbb{E}[\mu_2]$. We say that arm $i$ is *posterior-good* (resp., *posterior-bad*) after round $t$ if it has a larger (resp., smaller) posterior-mean reward $\mathbb{E}[\mu_i \mid H_t]$.

Among the two non-skip arms, the *good* (resp., *bad*) arm is the one with a larger (resp., smaller) mean reward. Clearly, the optimal per-episode policy given $\vec{\mu}$ is restricted to the good arm and the skip-arm (see Lemma A.2).

### 2.3. Additional Discussion

**Selection costs** in our model represents agent's time/effort for one more pull of a non-skip arm. They do *not* represent the full cost of exploration, *i.e.,* the opportunity cost of not exploiting. In substance, they ensure that (i) choosing an arm with a super-small reward should be strictly worse than not doing anything in this round, and so (ii) agents would not keep exploring under little/no incentive to do so. We believe this reflects the motivating applications.

**Full observability** of the history is crucial for our results, and it is often achieved by online platforms that aggregate user reviews. Note that aggregate statistics (the number of pulls and average reward for each arm) suffice for our purposes. Characterizing what happens when the history is only partially observed is an important open question which is not understood even for $m = 1$.

**Conjugate priors** are essential to our analysis, allowing for explicit posterior updates.[7] Beta-Bernoulli conjugate priors is a very standard modeling choice throughout all Bayesian analyses in the literature. Going beyond conjugate priors stretches the motivating story, as human agents are unlikely to engage in complex Bayesian reasoning.

**Early termination.** An agent can "skip" all remaining rounds, getting a reward of 0 and no cost for each skipped round. This corresponds to terminating the episode early for paradigmatic aggregation functions such as sum and max.

## 3. Learning Failures and Regret

This section discusses several notions of *learning failure* relevant to `EpiBSL` and our analysis and draws pertinent connections between them. (We establish existence of these failures in Section 4.)

A fundamental failure of exploration is when the good arm is not played more than a few times, *i.e.,* the agents give up on this arm before realizing that it is good. We use a slightly stronger notion to derive concrete implications for regret.

---

[7]We conjecture that our analysis should carry over to any bounded rewards and any priors under some mild assumptions (*e.g.,* the posterior should be monotone in realized rewards), but we don't know how to handle the technicalities.

**Definition 3.1.** Fix $c > 0$ and $N \in \mathbb{N}$. Event $\texttt{FAIL}_{c,N}$ occurs if the following three conditions hold:

- (Bounded mean rewards) $\mu_1, \mu_2 \in (c, 1 - c)$.

- (reward-gap) $\mu_2 \geq \mu_1 + c$

- (Bounded #pulls) Arm 2 is pulled at most $N$ times.

This is the type of learning failure that we establish going forward (as holding with constant probability). Crucially, we need the best arm is better by some margin. For technical convenience, we also include mean rewards being bounded and arm 2 being good (whereas arm 1 is prior-good).

A standard "end-to-end" notion of a learning failure is linear Bayesian regret, as a function of time. (Because *sublinear* Bayesian regret is a standard "minimal desiderata" for Bayesian bandit algorithms.) Deriving it from Definition 3.1 is somewhat non-trivial technically, requiring an intermediary: a version of Definition 3.1 which requires the good arm to not even be *considered* by the per-episodic policies.

**Definition 3.2.** A per-episodic policy *considers* an arm if it chooses this arm during the episode with positive probability. An episode $e$ considers an arm if the resp. policy $\pi_e$ does.

**Definition 3.3.** Fix $c > 0$ and $N \in \mathbb{N}$. Event $\texttt{StrongFail}_{c,N}$, called the *strong failure*, occurs if

- (*reward-gap*) $\mu_2 \geq \mu_1 + c$, and

- At most $N$ episodes consider arm 2.

We obtain the strong failure from $\texttt{FAIL}_{c,N}$ for any problem instance, as per the following lemma.

**Lemma 3.4.** *Fix $c \in (0, 1/2)$ and $N \in \mathbb{N}$. Consider an* EpiBSL *instance such that $\delta := \Pr\left[\texttt{FAIL}_{c,N}\right] > 0$. Then*

$$\Pr\left[\texttt{StrongFail}_{c,N'} \mid \texttt{FAIL}_{c,N}\right] \geq 1/2, \qquad (2)$$

*for some $N' < \mathbb{N}$ determined by $c, N, \delta$ and length $m$.*

*Proof Sketch.* As a shorthand, we say that a *good* episode is one that considers arm 2. On a high level, we want to argue that arm 2 is chosen more than $N$ times if there are sufficiently many good episodes, contradicting $\texttt{FAIL}_{c,N}$.

Making this argument formal requires some care. Condition on event $\mathcal{E}$ that $\mu_1, \mu_2 \in (c, 1 - c)$. Then in each good episode, arm 2 is chosen with some minimal probability $q > 0$ (determined by $c, m$). Choosing $N'$ large enough, define event $\mathcal{E}^*$ as follows: if there are at least $N'$ good episodes, then arm 2 is chosen more than $N$ times in the first $N'$ good episodes. Invoking Azuma-Hoeffding inequality, we obtain that $\Pr\left[\mathcal{E}^* \mid \mathcal{E}\right] \geq 1 - \delta/2$. Now, event $\texttt{FAIL}_{c,N} \cap \mathcal{E}^*$ immediately implies $\texttt{StrongFail}_{c,N'}$ and happens with probability at least $\delta/2$. This implies Eq. (2). $\qquad\square$

Now, let's focus on the strong failure and prove that it implies linear Bayesian regret. Most immediately, we get a statement in terms of (essentially) the gap in expected per-episode utility between the good arm and the bad arm, as per the following definition.

**Definition 3.5.** Fixing $\vec{\mu}$, the *utility-gap* is

$$G(\vec{\mu}) := U^*(\vec{\mu}) - \sup_{\pi \in \Pi_{\texttt{bad}}} V\left(\pi \mid \vec{\mu}\right) \geq 0,$$

where $\Pi_{\texttt{bad}}$ is the set of all per-episode policies that do not consider the good arm (but may skip some rounds).

**Theorem 3.6.** *Consider an instance of* EpiBSL. *If event* $\texttt{StrongFail}_{c,N}$ *holds, for some $c > 0$ and $N \in \mathbb{N}$, then*

$$\texttt{Reg}(T) \geq (T - N) \cdot G(\vec{\mu}) \quad \textit{for all episodes } T. \qquad (3)$$

*Remark* 3.7. Eq. (3) holds deterministically whenever the realized mean rewards $\vec{\mu}$ and the social dynamics satisfy the strong failure event. This follows directly from our definition of regret in Section 2.

The utility gap $G(\vec{\mu})$ is typically bounded away from $0$. Here's a concrete (even if somewhat inefficient) general statement to this extent, which lower-bounds $G(\vec{\mu})$ whenever the cost is sufficiently small.

**Lemma 3.8.** *Fix $c > 0$. Consider a realized problem instance of* EpiBSL *such that $\mu_2 \geq \mu_1 + c$. Then*

$$c_{\text{sel}} \leq \frac{c'}{2m} \quad \textit{implies} \quad G(\vec{\mu}) \geq c'/2,$$

*where $c' > 0$ is determined by $c$ and function $f$.*

*Proof Sketch.* The high level idea is as follows. We first ignore the effect of the cost, comparing policies only by the expected value of $f(.)$ under their execution, which we denote with $V'(\pi)$. Let $\pi_i$ denote the policy choosing arm $i$ for all rounds; we show that $V'(\pi_2) - V'(\pi_1) \geq c'$. To do this, let $\vec{r}_1, \vec{r}_2$ denote the rewards obtained from executing each of these two policies. Since $\mu_2 \geq \mu_1 + c$, we can couple the rewards in a way that $\vec{r}_1$ is always coordinate-wise larger than or equal to $\vec{r}_2$. Since $f(.)$ is coordinate-wise non-decreasing, this means that with probability 1 we have $f(\vec{r}_2) \geq f(\vec{r}_1)$. Moreover, using $\mu_2 \geq \mu_1 + c$, we show that with some strictly positive probability (depending only on $c, f$) we have $\vec{r}_2 = \vec{1}$ and $\vec{r}_1 = \vec{0}$. Since $f(\vec{1}) > f(\vec{0})$, taking expectation we conclude that $V'(\pi_2) - V'(\pi_1) \geq c'$ for some suitable $c'$ depending only on $c, f$.

The above argument compares $\pi_1$ and $\pi_2$ which attain the maximum value for $V'(\pi)$ when restricting, respectively, to policies not choosing arm 2 and policies not choosing arm 1. When we look at $V(\pi)$ however, both policies may be sub-optimal and both the optimal policy and the policy maximizing $V(\pi)$ in $\Pi_{\texttt{bad}}$ may in principle use the skip arm. To address this, we use the bound on cost to show that

that the gain obtained from skipping is smaller than the gap between $V'(\pi_2) - V'(\pi_1)$. Formally, $c_{\text{sel}} \leq c'/2m$ ensures that for all $\pi$ we have $|V(\pi) - V'(\pi)| \leq c'/2$. This allows us to lower bound $U^*$ with $V'(\pi_2) - c'/2$ and upper bound $\sup_{\pi \in \Pi_{\text{bad}}} V(\pi)$ with $V'(\pi_1)$. For the latter bound, we are relying on the fact that $\pi_1$ maximizes $V'(\pi)$ in $\Pi_{\text{bad}}$. Taking the difference finishes the proof. $\qquad\square$

While the previous lemma provides a general lower bound on $G(\vec{\mu})$ which is fairly opaque, one can get more lucid bounds for specific functions $f$. For example:

**Lemma 3.9.** *If $f$ is the $\min$ function then*

$$G(\vec{\mu}) \geq \max\left\{0, \, |\mu_2^m - \mu_1^m| - m\, c_{\text{sel}}\right\}.$$

*If $f$ is the $\max$ function then*

$$G(\vec{\mu}) \geq \max\left\{0, \, |(1-\mu_1)^m - (1-\mu_2)^m| - m\, c_{\text{sel}}\right\}.$$

Note that the lower bounds in Lemma 3.9 are quite strong for small $m$ and "most" $\vec{\mu}$ values.

# 4. Failure Results and Techniques

This section spells out our results on existence and prevalence of exploration failures, and the respective implications on Bayesian regret. Throughout, we focus on the failure event $\texttt{FAIL}_{c,N}$ from Definition 3.1 and guarantee that it happens with positive probability for some prior-dependent parameters $c, N$. We guarantee this as a common case: for any prior $\mathcal{P}$ and any aggregation function $f$, as they are defined in Section 2.

Our main result allows arbitrary episode length $m \geq 2$.

**Theorem 4.1.** *Consider an* $\texttt{EpiBSL}$ *instance with an arbitrary episode length $m \geq 2$ and symmetric function $f$. Then the following property holds:*

*(P1) There exist constants $N_{\mathcal{P}} < \infty$ and $c_{\mathcal{P}}, q_{\mathcal{P}} > 0$ depending only on prior $\mathcal{P}$, and constant $c_{\max} > 0$ depending only $\mathcal{P}$ and $f$, such that small enough cost $c_{\text{sel}} < c_{\max}$ implies*

$$\Pr\left[\texttt{FAIL}_{c_{\mathcal{P}}, N_{\mathcal{P}}}\right] \geq q_{\mathcal{P}}^m > 0.$$

*Remark* 4.2. Here, $N_{\mathcal{P}} = 1 + \lfloor (m-1)\beta_2/\alpha_2 \rfloor$.

Next, we focus on length $m = 2$ and remove the symmetry assumption in Theorem 4.1.

**Theorem 4.3.** *Consider an* $\texttt{EpiBSL}$ *instance with episode length $m = 2$ and not (necessarily) symmetric function $f$. Then Property (P1) holds.*

*Remark* 4.4. To make the parameter choice more explicit,

$$N_{\mathcal{P}} = 2 + \left\lfloor \frac{\beta_2}{\alpha_2} \right\rfloor + \left\lfloor \frac{4}{3} \cdot \frac{\mathbb{E}\left[\mu_2\right]}{\mathbb{E}\left[\mu_1 - \mu_2\right]} \right\rfloor;$$

$$c_{\max} = \frac{\mathbb{E}\left[\mu_1\right] - \mathbb{E}\left[\mu_1 - \mu_2\right]/4}{f(1,1) - f(1,0)}$$

if $f(1,1) > f(1,0)$. We can take $c_{\max} = 1$ otherwise.

To motivate non-symmetric aggregation functions $f$, one example is $f(r_1, r_2) = r_2$. Here, the first round is a "trial run", and the second round determines the score. Another example is linear functions: $f(r_1, r_2) = a \cdot r_1 + b \cdot r_2$ with $a \neq b$, so the two rounds do not matter equally.

Leveraging the machinery developed in the previous section, we obtain linear Bayesian regret for problem instances covered by Theorems 4.1 and 4.3.

**Corollary 4.5.** *Assume $m = 2$ or that $f$ is symmetric. Then for some constants $c_0 > 0$ and $N_0, c_{\max} < \infty$ determined by the problem parameters $(f, \mathcal{P}, m)$, small enough cost $c_{\text{sel}} < c_{\max}$ implies linear Bayesian regret:*

$$\texttt{BReg}(T) \geq c_0 \cdot (T - N_0) \qquad \text{for any episode } T.$$

*Proof Sketch.* We start from the failure events guaranteed by Theorems 4.1 and 4.3 and consecutively apply Lemma 3.4, Theorem 3.6, and Lemma 3.8. $\qquad\square$

Finally, we achieve Property (P1) without an upper bound on cost, for some special cases.

**Theorem 4.6.** *Consider an* $\texttt{EpiBSL}$ *instance. Assume either that (a) $m = 2$ and function $f$ is symmetric, or that (b) function $f$ is specifically the $\max$ or the $\min$. Then Property (P1) holds with $c_{\max} = 1$.*

*Remark* 4.7. Here, $N_{\mathcal{P}} = 1 + \lfloor \beta_2/\alpha_2 \rfloor$.

We do not provide a new regret corollary since our current techniques would still require an upper bound on the cost, akin to Corollary 4.5.

## 4.1. Proof Sketches for the Theorems

We sketch out the proofs of Theorems 4.1, 4.3 and 4.6 in what follows. The logic proceeds in the opposite order: we start with Theorem 4.6(a) as the base case, then refine the argument to obtain the other results.

Analyses from prior work do not directly carry over. Indeed, Banihashem et al. (2023); Slivkins et al. (2025) focus on single-round episodes, and analyses therein focus on comparing *current* posterior-mean rewards (or current sample-average rewards, for non-Bayesian models). In $\texttt{EpiBSL}$, however, an agent may rationally select a posterior-bad arm, since a success early in the episode can substantially improve its posterior and make it preferable in later rounds.

This phenomenon is illustrated via a simple example in Appendix G.1. This within-episode effect is present even for length $m = 2$ and simple symmetric utilities.

Instead, we reason about *optimistic* posteriors of a given arm: ones that would be induced by hypothetical *positive* future observations of this arm in the same episode.

Our proofs follow a common template. We are interested in the failure event when arm 2 is the good arm by some margin *and* it is not chosen except in a few episodes. To this end, we identify events under which arm 1's posterior remains sufficiently strong, while arm 2 accumulates enough unfavorable evidence so that — even under optimistic posteriors — it is never selected by any future agent. The main technical difficulty is controlling the trajectories of optimistic posteriors and understanding how these posteriors interact with episode length, selection costs, and the structure of the utility function.

**Theorem 4.6(a)** targets the simplest episodic setting, with episode length $m = 2$ and symmetric function $f$, which already exhibits the challenges outlined above.

The first key ingredient is the *bad-start condition* on the arms' posteriors: a condition that forces an agent to not use the good arm in the first round of the episode. [8] Essentially, this condition needs to be robust to optimistic posteriors. A sufficient bad-start condition is that arm 2 is *strongly posterior-bad*: posterior-bad currently and (hypothetically) even after $m - 1$ more positive samples from this arm. This is established in Lemma C.1.

Second, we isolate the behavior of arm 1 via an event under which its posterior mean never drops too far below $\mathbb{E}[\mu_1]$, call this event *arm1-stability*. Lemma C.2 and Lemma C.3 show that this event occurs with constant probability over the prior, using a stopping-time and martingale argument.[9] Conditioned on this event, we can guarantee that if arm 2 starts out as strongly posterior-bad, it remains so while it is not played (and, in particular, this arm is not chosen in the second round of the episode).

Unlike single-round analyses in prior work, we cannot argue that arm 2 is never chosen by the agents. Instead, we focus on the event that the first few samples of this arm are all zeroes.[10] Then Lemma C.4 shows that the optimistic posterior mean reward of arm 2 is driven permanently below arm 1's posterior (under arm1-stability), after which Lemma C.1 prevents any further pulls. Combining these

_______________

[8]We focus on the first round of the episode, as this suffices for our argument. Our condition alone does not preclude arm 2 from being played in the second round.

[9]This particular argument is similar to the one used in Banihashem et al. (2023) for the non-episodic case ($m = 1$).

[10]It would have sufficed to consider a short prefix with abnormally low average reward.

events with the realization of the mean rewards such that $\mu_2$ is significantly larger than $\mu_1$ yields the failure event with constant probability.

*Remark* 4.8. We need to follow the same proof outline even if function $f$ is simple — max or sum — but proving Lemma C.1 simplifies somewhat.

**Theorem 4.1 (longer episodes, symmetric utilities).** We show that "arm 2 is strongly posterior-bad" is a bad-start condition. However, we need to use induction on episode length $m$ to handle the additional rounds. (As before, arm1-stability ensures that the property of arm 2 being strongly posterior-bad is preserved while arm 2 is not played.)

The inductive argument proceeds as follows. Suppose, for the sake of contradiction, that arm 2 is strongly posterior-bad at the start of an episode, and yet an agent starts by choosing this arm. Even if arm 2 yields a reward of 1, the remaining interaction corresponds to a shorter episode of length $m - 1$, and arm 2 is strongly posterior-bad for length $m - 1$. Therefore, by the inductive hypothesis, arm 2 will not be selected in the "first" round of the remaining episode (*i.e.,* the second round of the original episode). Thus, for the second round, the policy must either play arm 1 or skip.

If agent's policy always plays arm 1 in the second round, symmetry of function $f$ implies that this policy is equivalent in expectation to a policy that swaps the first two rounds (playing arm 1 in round 1 and arm 2 in round 2). Such a policy cannot be posterior-optimal, however. This is because — even if arm 1 gives reward 1 in the first round — in the remaining episode, arm 2 is again strongly posterior-bad for length $m - 1$, so by the inductive hypothesis arm 2 cannot be chosen in round 2.

For $m > 2$, a policy may in principle admit a richer set of behaviors, including decisions that depend on observed rewards and potentially involve skipping. To keep the analysis within a tractable inductive framework, we impose an additional assumption that the selection cost is sufficiently small. This assumption is used to restrict the class of policies that need to be considered — ensuring that never skipping is strictly preferred in expectation if at all possible (*i.e.,* if receiving positive rewards for the rest of the episode strictly maximizes $f$). This allows the inductive argument to go through. We refer to Lemma D.3 for more details.

**Theorem 4.3 (general utilities, $m = 2$).** Theorem 4.3 highlights a subtle issue that does not arise for symmetric utilities: "arm 2 being strongly posterior-bad" no longer suffices as a bad-start condition.

The issue is as follows. Call arm $i$ *hopeful* in round $t$ if it can become posterior-good given one sample of some arm (*i.e.,* either with one positive sample of the same arm, or with one negative sample of the other non-skip arm). While

it may seem that the bad-start condition causes arm 2 to be not hopeful, this is not the case: it is possible that sampling from arm 1 and observing reward 0 causes the posterior mean of arm 1 to drop below that of arm 2. This issue is important for functions where, intuitively, the second round matters more and, crucially, it only matters more when the first reward is 0. In these cases, choosing arm 1 in the first round can be "risky" since, if get a reward of 0, neither of the two arms have a high posterior for the second round. A concrete example of this is provided in Appendix G.2 where we show that the agent should first choose arm 2 and then choose arm 1 if the reward was 0 and choose skip otherwise.

We deal with this issue by assuming an upper bound on the cost similar to Theorem 4.1, ensuring that the skip arm is not preferred unless a reward of 1 in the second round does not increase $f$ (in which case it is trivially preferred). The assumption allows us to characterize functions for which skip is chosen in the second round, showing that examples such as the one in Appendix G.2 are not possible.

## 5. Discussion and Extensions

**Two arms and beyond.** Our paper is motivated by Question (Q) in the Introduction and resolves it in the negative. While even one counterexample rules out a general positive result, what we proved is much stronger: learning failure is ubiquitous for two arms. Therefore, "learning success" for $K > 2$ arms, if any, must be confined to some exotic regimes which completely exclude our 2-armed setting.

We emphasize that idealized models, and particularly ones limited to two arms, are very common in closely related work, *e.g.,* the work on "strategic experimentation" and "incentivized exploration" referenced in Section 1.1, as well as the literature on "sequential social learning" (surveyed in Golub & Sadler, 2016).

That said, our analysis easily extends to a broad family of problem instances with $K > 2$ arms, where two arms look much better than all others according to the prior. (The sufficient condition for this, Equation (13), depends only on the prior and $m$.) Our analysis deals with some events for these two arms that cause a failure, and under these events all other arms are never chosen. We spell out the results and the required modifications to the analysis in Appendix H.

**Costs/skips and beyond.** Our model with costs and skips reflects the motivating applications, ensuring that the agents would not keep exploring under little/no incentive to do so.

That said, our analysis sheds light on the model with $c_{\text{sel}} = 0$. For concreteness, focus on $m = 2$ and symmetric aggregation function $f$. Suppose skips are allowed and ties are broken in favor of skips. Then our analysis carries over with (only) minor modification, see Appendix I.

Without skips, selection costs don't matter (because each agent incurs the same cost per round no matter what), and our failure analysis seems to crumble. In fact, we have "learning success" for $f = \max$ (resp., $f = \min$) under any random tie-breaking. Indeed, if round 1 of an episode returns reward 1 (resp., reward 0), then the subsequent round(s) does not affect agent's utility. So, we have a tie, and each arm gets chosen with some probability.

**Episode length.** Our lower bound on failure probability decays exponentially in episode length $m$ (see Theorem 4.1). The small-$m$ regime, in which this exponential decay is relatively insignificant, captures realistic scenarios: *e.g.,* try a few prompts or products. How the *actual* failure probability scales with $m$ is an open question.

In general, quantitatively strong lower bounds on failure probability of the "greedy" bandit algorithm are hard to come by: essentially, they are only known for two-armed bandits (Banihashem et al. (2023), *i.e.,* the special case of our model with $m = 1$). But even a tiny-but-positive failure probability is important as it implies linear regret, the basic notion of failure in bandit learning.

## Impact Statement

Our goal is to advance theoretical understanding of machine learning. We do not foresee any immediate societal consequences of our work. Longer-term consequences, if any, will be cumulative over a large body of machine learning theory. While this point applies generically across most/all of learning theory, one potential implication specific to this paper is to emphasize the need for exploration in bandit-like social-learning dynamics, even when individual agents may explore for themselves.

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

# APPENDICES

# A. Additional Preliminaries

**Notation.** Throughout the proofs, we use $\gamma_{i,t}$ to denote $\frac{\alpha_{i,t}}{\alpha_{i,t}+\beta_{i,t}}$. We also frequently use $\alpha_{i,0}, \beta_{i,0}$ to denote the initial prior from which $\mu_i$ is drawn. Note that $\gamma_{i,0} = \mathbb{E}[\mu_i]$. We use $\Delta := \gamma_{1,0} - \gamma_{2,0}$ to denote the gap between the prior mean of the arms 1 and 2. We additionally use $\Theta_{\mathcal{P}}(.)$ to hide constants under the prior; e.g., $\Theta_{\mathcal{P}}(1)$ denotes some constant that depends on the prior $\mathcal{P}$.

Given a policy $\pi$ and an episode $e$, we use $\mathtt{PU}_\pi = \mathbb{E}\left[U(\pi_e) \mid \widetilde{H}_e\right]$ to denote its *Posterior Utility* which we define as the expected utility of the policy conditioned on the history at the start of the episode. As previously mentioned, we assume that the agent chooses some policy maximizing $\mathtt{PU}_\pi$. We refer to any such policy as a *posterior optimal* policy.

**Reward Tapes.** For analytical clarity we represent rewards via *reward tapes*. At the start of the horizon an infinite sequence $(\mathtt{tape}_{i,1}, \mathtt{tape}_{i,2}, \ldots)$ is drawn for every arm $i$. Conditional on $\mu_i$, the entries are i.i.d. Bernoulli$(\mu_i)$ and are independent across arms. Whenever arm $i$ is pulled for the $\ell$-th time, the agent observes the next unrevealed entry $\mathtt{tape}_{i,\ell}$. In this view, all randomness is determined upfront; the only uncertainty that remains is which tape entries are revealed by the sequence of pulls. This perspective is commonly used in the bandit literature.

We use $n_{a,t}$ to denote the number of times that arm $a$ has been pulled from rounds 1 to $t$ (inclusive). We define $\alpha_a^{(n)}$ to be the posterior value of $\alpha_a$ if we were to see $\mathtt{tape}_{a,1}, \ldots, \mathtt{tape}_{a,n}$; i.e., $\alpha_a^{(n)} = \alpha_{a,0} + \sum_{\ell \le n} \mathtt{tape}_{a,\ell}$. Note that $\alpha_{a,t} = \alpha_a^{(n_{a,t})}$. The values $\beta_a^{(n)}$ and $\gamma_a^{(n)}$ are defined similarly.

The environment first draws $\mu_i$ and then draws $(\mathtt{tape}_{i,1}, \ldots, \mathtt{tape}_{i,2}, \ldots)$ for each $i$ from Bernoulli$(\mu_i)$. Since the algorithm never actually sees $\mu_i$ however, for the purposes of our analysis it is easier to view it as a hidden variable and sample $\mathtt{tape}_{i,\ell}$ directly. Concretely, we assume that $\mathtt{tape}_{i,1}$ is drawn from the distribution Bernoulli$(\gamma_{i,0})$ and after sampling $\mathtt{tape}_{i,1}, \ldots, \mathtt{tape}_{i,\ell-1}$, the value $\mathtt{tape}_{i,\ell}$ is drawn from the distribution Bernoulli$(\gamma_i^{(\ell-1)})$. Note that this perspective is correct because, conditioned on $\mathtt{tape}_{i,1}, \ldots \mathtt{tape}_{i,\ell-1}$, the random variable $\mu_i$ has the distribution Beta$(\alpha_i^{(\ell-1)}, \beta_i^{(\ell-1)})$ which means

$$\Pr\left[\mathtt{tape}_{i,\ell} = 1 \mid \mathtt{tape}_{i,1}, \ldots \mathtt{tape}_{i,\ell-1}\right] = \mathbb{E}_{\mu_i \sim \mathrm{Beta}(\alpha_i^{(\ell-1)}, \beta_i^{(\ell-1)})}\left[\Pr_{\mathtt{tape}_{i,\ell} \sim \mathrm{Bernoulli}(\mu_i)}\left[\mathtt{tape}_{i,\ell} = 1\right]\right]$$
$$= \mathbb{E}_{\mu_i \sim \mathrm{Beta}(\alpha_i^{(\ell-1)}, \beta_i^{(\ell-1)})}[\mu_i] = \gamma_i^{(\ell-1)}.$$

**Concentration.** We use the following standard concentration inequality which follows from Azuma–Hoeffding.

**Lemma A.1.** *Fix $c, \delta > 0$ and $n \in \mathbb{N}$. Consider a sequence of random variables $(\nu_1, X_1; \ldots; \nu_n, X_n)$ realized in this order, such that each $X_t$, $t \in [n]$, is an independent Bernoulli draw with mean $\nu_t \in [c, 1]$. If $n_0 \ge \frac{8}{c^2}\log(1/\delta)$, then the sum $S_n = \sum_{t \in [n]} X_i$ satisfies $\Pr[S_n \ge cn/2] \ge 1 - \delta$.*

**Optimal policy.** The following lemma shows that the optimal policy never chooses the bad arm.

**Lemma A.2.** *For any $\vec{\mu}$ with $\mu_1 > \mu_2$, the optimal per-episode policy does not consider arm 2.*

*Proof.* We will show that any deterministic policy $\pi$ considering arm 2 has a strictly lower expected utility than some randomized policy. Since any randomized policy is itself a distribution over deterministic policies, this would imply that there must exist some deterministic policy with expected utility higher than $\pi$.

We proceed with a proof. Consider the policy $\pi'$ that, for each history $h$ such that $\pi$ chooses arm 2, randomizes between the skip arm and arm 1 by playing arm 1 with probability $\frac{\mu_2}{\mu_1}$ and skip with probability $1 - \frac{\mu_2}{\mu_1}$. For histories for which $\pi$ chooses either arm 1 or skip, $\pi'$ chooses the same arm as $\pi$.

It is clear that the probability of observing a reward 1 using this approach is $\frac{\mu_2}{\mu_1} \cdot \mu_1 = \mu_2$. As such, the reward sequence obtained by $\pi'$ has the same distribution as the reward sequence obtained by $\pi$. Importantly however, the expected cost incurred by $\pi'$ is strictly lower. Concretely, for any input, either the two policies have the same output, in which case they incur the same cost, or $\pi$ chooses arm 2, in which case $\pi$ pays the cost $c_{\mathrm{sel}}$ while $\pi'$ pays, in expectation, the cost $\frac{\mu_2}{\mu_1}c_{\mathrm{sel}}$. Since the reward distributions of $\pi$ and $\pi'$ are the same, this implies that the expected cost incurred by $\pi'$ is not higher than that of $\pi'$. Additionally, since $\pi$ chooses arm 2 with positive probability over the draws of the arms, the inequality is strict. It follows that $V(\pi') > V(\pi)$, finishing the proof. $\square$

# B. Proofs for Section 3: Failures and Regret

## B.1. Proof of Lemma 3.4: from $\texttt{FAIL}_{c,N}$ to strong failure

Fix the algorithm and reveal randomness episode by episode. Let $\pi_e$ denote the per-episode policy used in episode $e$, which is fully determined by the history before episode $e$.

Let $N'$ be an integer to be determined later. We want to show that if sufficiently many episodes consider arm 2 then arm 2 will be chosen $N$ times. To prove this, we define a sequence $(\widetilde{Y}_1, \ldots, \widetilde{Y}_{N'}) \in \{0, 1\}^{N'}$ as follows. Let $E_j$ denote the $j$-th episode that considers arm 2 (if it exists). Set $\widetilde{Y}_j = 1$ if $E_j$ does not exist or arm 2 is chosen in episode $E_j$, and $\widetilde{Y}_j = 0$ otherwise. (Setting $\widetilde{Y}_j = 1$ when $E_j$ does not exist is a convention that only strengthens the lower bound on $\sum_j \widetilde{Y}_j$.)

Let $\mathcal{E}$ denote the event that $\mu_1, \mu_2 \in (c, 1 - c)$. Since the episodes are considered sequentially, the values $\widetilde{Y}_j$ are realized in order. We claim that, conditioned on $\mathcal{E}$, whenever $\widetilde{Y}_j$ is realized, we have $\mathbb{E}\left[\widetilde{Y}_j\right] \geq c^m$. To see why, consider two cases.

- If the reason the value is being realized is we had at most $j - 1$ episodes considering arm 2, then the value is set to 1 and the claim holds.

- Otherwise, the value $\widetilde{Y}_j$ is being realized because in some episode $e$ the policy $\pi_e$ considers arm 2. By definition, there exists a within-episode reward realization of length at most $m$ that causes $\pi_e$ to play arm 2. Because $\mu_1, \mu_2 \in (c, 1 - c)$, any fixed such realization occurs with probability at least $c^m$.

Let $\texttt{Bridge}_{N'}$ denote the event $\left\{\sum_{j=1}^{N'} \widetilde{Y}_j \geq N + 1\right\}$. We claim that if

$$N' \geq \max\left\{\frac{8}{c^{2m}} \log \frac{2}{\delta}, \frac{2}{c^m}(N + 1)\right\},$$

then

$$\Pr\left[\texttt{Bridge}_{N'} \mid \mathcal{E}\right] \geq 1 - \delta/2. \tag{4}$$

To prove this, apply Lemma A.1 to $(\widetilde{Y}_j)_{j=1}^{N'}$ with $c = c^m$ and parameter $\delta/2$. Since $N' \geq \frac{8}{c^{2m}} \log \frac{2}{\delta}$, we have

$$\Pr\left[\sum_{j=1}^{N'} \widetilde{Y}_j \geq \frac{c^m}{2} N'\right] \geq 1 - \delta/2.$$

Since $\frac{c^m}{2} N' \geq N + 1$, this implies that with probability at least $1 - \delta/2$,

$$\sum_{j=1}^{N'} \widetilde{Y}_j \geq N + 1,$$

finishing the proof of Equation (4).

We next show that

$$\texttt{FAIL}_{c,N} \cap \texttt{Bridge}_{N'} \subseteq \texttt{StrongFail}_{c,N'}.$$

To see why, assume $\texttt{FAIL}_{c,N}$ and $\texttt{Bridge}_{N'}$ both occur. If at least $N'$ episodes consider arm 2, then the $\widetilde{Y}_j$ values equal to 1 correspond to episodes where arm 2 was pulled and as such $\texttt{Bridge}_{N'}$ implies that arm 2 was pulled more than $N$ times in total, contradicting $\texttt{FAIL}_{c,N}$. Therefore, under $\texttt{FAIL}_{c,N}$ we must have fewer than $N'$ episodes that consider arm 2, which is exactly the structural condition in $\texttt{StrongFail}_{c,N'}$.

Since $\Pr\left[\texttt{FAIL}_{c,N}\right] \geq \delta$ by assumption, letting $\texttt{Bridge}_{N'}^c$ denote the complement event of $\texttt{Bridge}_{N'}^c$, we have

$$\Pr\left[\texttt{StrongFail}_{c,N'} \mid \texttt{FAIL}_{c,N}\right]$$
$$\geq \Pr\left[\texttt{Bridge}_{N'} \mid \texttt{FAIL}_{c,N}\right]$$
$$= 1 - \Pr\left[\texttt{Bridge}_{N'}^c \mid \texttt{FAIL}_{c,N}\right]$$
$$= 1 - \frac{\Pr\left[\texttt{Bridge}_{N'}^c \cap \texttt{FAIL}_{c,N}\right]}{\Pr\left[\texttt{FAIL}_{c,N}\right]}$$

$$\geq 1 - \frac{\Pr\left[\texttt{Bridge}_{N'}^c \cap \mathcal{E}\right]}{\Pr\left[\texttt{FAIL}_{c,N}\right]}$$
$$\geq 1 - \frac{\delta/2}{\delta} = \tfrac{1}{2},$$

which completes the proof.

### B.2. Proof of Lemma 3.8: utility-gap

Define $\pi_1'$ and $\pi_2'$ to be the policies that pull arm 1 and arm 2, respectively, in all $m$ rounds and never skip. We begin by comparing their cost-free utilities.

Sample $\vec{r}^{(1)} = (r_1^{(1)}, \ldots, r_m^{(1)})$ with independent coordinates distributed as $\texttt{Bernoulli}(\mu_1)$. Conditioned on $\vec{r}(1)$, construct $\vec{r}(2) = (r_1^{(2)}, \ldots, r_m^{(2)})$ as follows. If $r_i^{(1)} = 1$, set $r_i^{(2)} = 1$. If $r_i^{(1)} = 0$, set $r_i^{(2)}$ to 0 with probability $(1 - \mu_2)/(1 - \mu_1)$ and to 1 otherwise. Since $\mu_2 \geq \mu_1 + c$, this ensures that each coordinate of $\vec{r}(2)$ is distributed as $\texttt{Bernoulli}(\mu_2)$; this holds because $\Pr\left[r_i^{(2)} = 0\right] = \Pr\left[r_i^{(1)} = 0\right]\Pr\left[r_i^{(2)} = 0 \mid r_i^{(1)} = 0\right] = (1 - \mu_1)\frac{1 - \mu_2}{1 - \mu_1} = 1 - \mu_2$. Moreover, we always have $\vec{r}(2) \geq \vec{r}(1)$ coordinate-wise.

Because $f$ is coordinatewise increasing and $\vec{r}(2) \geq \vec{r}(1)$ coordinate-wise, the random variable $f(\vec{r}(2)) - f(\vec{r}(1))$ is always non-negative. We now lower bound the probability that it is bounded below by a constant positive. Since $\mu_2 \geq \mu_1 + c$, we have $1 - \mu_1 \geq 1 - \mu_2 + c \geq c$, and therefore

$$\Pr\left[\vec{r}^{(1)} = \vec{0}\right] = (1 - \mu_1)^m \geq c^m.$$

Conditioned on this event, each coordinate of $\vec{r}(2)$ flips to 1 with probability $1 - \frac{1 - \mu_2}{1 - \mu_1} = \frac{\mu_2 - \mu_1}{1 - \mu_1} \geq c$, independently across coordinates. Thus,

$$\Pr\left[\vec{r}^{(2)} = \vec{1} \mid \vec{r}(1) = \vec{0}\right] \geq c^m.$$

Hence, with probability at least $c^{2m}$, we have simultaneously $\vec{r}(1) = \vec{0}$ and $\vec{r}(2) = \vec{1}$. On this event,

$$f(\vec{r}(2)) - f(\vec{r}(1)) = f(\vec{1}) - f(\vec{0}).$$

Let $V'(\pi)$ denote the expected value of $f(\vec{r})$ when using policy $\pi$. Since we always have $f(\vec{r}(2)) - f(\vec{r}(1))$, taking expectation we obtain

$$V'(\pi_2') - V'(\pi_1') = \mathbb{E}[f(\vec{r}(2)) - f(\vec{r}(1))]$$
$$\geq c^{2m} \cdot \left(f(\vec{1}) - f(\vec{0})\right).$$

Define

$$c' := c^{2m} \cdot \left(f(\vec{1}) - f(\vec{0})\right),$$

which depends only on $c$ and $f$.

We now relate cost-free and actual utilities. For any $m$-round policy $\pi$, the total incurred cost is non-negative and at most $m \cdot c_{\text{sel}}$. Therefore

$$V'(\pi) - m \cdot c_{\text{sel}} \leq V(\pi) \leq V'(\pi).$$

Since $f$ is coordinatewise increasing and $V'$ ignores costs, skipping cannot increase the score, and hence

$$V'(\pi_1^*) \leq V'(\pi_1').$$

By optimality of $\pi_2^*$ among policies using only arm 2 and skip, we also have $V(\pi_2^*) \geq V(\pi_2')$. Combining these inequalities,

$$
\begin{aligned}
V(\pi_2^*) - V(\pi_1^*) &\geq V(\pi_2') - V'(\pi_1') \\
&\geq \left(V'(\pi_2') - m \cdot c_{\text{sel}}\right) - V'(\pi_1') \\
&= \left(V'(\pi_2') - V'(\pi_1')\right) - m \cdot c_{\text{sel}} \\
&\geq c' - m \cdot c_{\text{sel}}.
\end{aligned}
$$

Thus, whenever $2m \cdot c_{\text{sel}} \leq c'$, we obtain

$$V(\pi_2^*) - V(\pi_1^*) \geq c'/2,$$

completing the proof.

### B.3. Lower bounds on utility-gap in special cases: proof of Lemma 3.9

Let's restate the lemma for convenience.

**Lemma B.1.** *Assume $\mu_2 \geq \mu_1$. If $f$ is the $\min$ function then*

$$G(\vec{\mu}) \geq \max\{0, \mu_2^m - \mu_1^m - mc_{\text{sel}}\},$$

*and if $f$ is the $\max$ function then*

$$G(\vec{\mu}) \geq \max\{0, (1 - \mu_1)^m - (1 - \mu_2)^m - mc_{\text{sel}}\}.$$

*Proof.* As in the proof of Lemma 3.8, let $V'(\pi)$ denote the expected value of $f(\cdot)$ under policy $\pi$ and let $\pi_i$ denote the policy choosing arm $i$ in all rounds. We first observe that $V'(\pi_i) = \mu_i^m$ for $f = \min$ and $V'(\pi_i) = 1 - (1 - \mu_i)^m$ for $f = \max$ since $\min(\vec{r}) = \mathbb{1}\left\{\vec{r} = \vec{1}\right\}$ and $\max(\vec{r}) = \mathbb{1}\left\{\vec{r} \neq \vec{0}\right\}$. It follows that $V'(\pi_2) - V'(\pi_1)$ is lower bounded by $\mu_2^m - \mu_1^m$ when $f$ is $\min$ and by $(1 - \mu_1)^m - (1 - \mu_2)^m$ when $f$ is $\max$.

Next, as in Lemma 3.8, we observe that since $|V(\pi) - V'(\pi)| \leq mc_{\text{sel}}$ we have $U^* \geq V(\pi_2) \geq V'(\pi_2) - mc_{\text{sel}}$ and $\sup_{\pi \in \Pi_{\text{bad}}} V(\pi) \leq \sup_{\pi \in \Pi_{\text{bad}}} V'(\pi) = V'(\pi_1)$. As before, taking the difference finishes the proof. Note that the bound $G \geq 0$ always holds by definition of $G$. $\qquad\square$

## C. Proof of Theorem 4.6(a): $m = 2$, Arbitrary Cost

In this section, we prove Theorem 4.6. We first establish the following lemma that provides a sufficient condition for arm 1 being chosen over arm 2.

**Lemma C.1.** *If $\frac{\alpha_{2,t-1}+1}{\alpha_{2,t-1}+\beta_{2,t-1}+1} < \gamma_{1,t-1}$, then arm 2 is not selected in round $t$.*

We defer the proof of the above lemma to Appendix C.1.

Define the event $\text{Ev}_1$ as

$$\text{Ev}_1 = \left\{ \frac{\alpha_{1,0} + \sum_{\ell=1}^n \text{tape}_{1,\ell}}{\alpha_{1,0} + \beta_{1,0} + n} \geq \gamma_{1,0} - \Delta/4 \quad \text{for all } n \in [1, T] \right\}. \tag{5}$$

This corresponds to the arm1-stability event defined in Section 4.1.

**Lemma C.2.** *If $\text{Ev}_1$ occurs, then for all $t \geq 0$, we have*

$$\gamma_{1,t} \geq \gamma_{1,0} - \Delta/4.$$

*Proof.* Let $n_{1,t}$ be the number of pulls of arm 1 up to (and including) round $t$. By the posterior update rule for Beta-Bernoulli distributions, we have

$$\gamma_{1,t} = \frac{\alpha_{1,0} + \sum_{\ell=1}^{n_{1,t}} \texttt{tape}_{1,\ell}}{\alpha_{1,0} + \beta_{1,0} + n_{1,t}}.$$

By definition of $\texttt{Ev}_1$, this quantity is always at least $\gamma_{1,0} - \Delta/4$. □

We next prove the following lemma which is similar to Theorem 7.1 in (Banihashem et al., 2023) and follows using the same argument. For completeness, we provide a proof.

**Lemma C.3.** $\Pr[\texttt{Ev}_1] \geq \Delta/4$.

*Proof.* Consider the hypothetical game where we just keep playing arm 1 and see its rewards. Observe that after playing $n$ rounds, the posterior value $\gamma_{1,n}$ will equal $\frac{\alpha_{1,0} + \sum_{\ell=1}^{n} \texttt{tape}_{1,\ell}}{\alpha_{1,0} + \beta_{1,0} + n}$. Define the stopping time $t_{\text{stop}}$ as the first time when $\gamma_{1,t}$ drops below $\gamma_{1,0} - \Delta/4$ and as $T+1$ otherwise. Observe that $t_{\text{stop}}$ is a stopping time since whether or not $t_{\text{stop}} = t$ depends only on $\texttt{tape}_{1,1}, \ldots, \texttt{tape}_{1,t}$. We further note that $\gamma_{1,t}$ is a martingale. Specifically, since we assumed we are only playing arm 1, we have $\gamma_{1,t} = \mathbb{E}\left[\mu_1 | \texttt{tape}_{1,1}, \ldots, \texttt{tape}_{1,t-1}\right]$. As such, $\gamma_{1,t}$ is a Doob martingale.

Therefore, $\mathbb{E}\left[\gamma_{1,t_{\text{stop}}}\right] = \gamma_{1,0}$. Observe however that

$$
\begin{aligned}
\gamma_{1,0} &= \mathbb{E}\left[\gamma_{1,t_{\text{stop}}}\right] \\
&= \Pr[t_{\text{stop}} = T+1]\mathbb{E}\left[\gamma_{1,t_{\text{stop}}} \mid t_{\text{stop}} = T+1\right] + \Pr[t_{\text{stop}} < T+1]\mathbb{E}\left[\gamma_{1,t_{\text{stop}}} \mid t_{\text{stop}} < T+1\right] \\
&\leq \Pr[t_{\text{stop}} = T+1] \cdot 1 + \Pr[t_{\text{stop}} < T+1](\gamma_{1,0} - \Delta/4), \quad\quad (i) \\
&\leq \Pr[t_{\text{stop}} = T+1] \cdot 1 + (\gamma_{1,0} - \Delta/4),
\end{aligned}
$$

where for $(i)$ we have used $\gamma_{1,t} \leq 1$ and definition of $t_{\text{stop}}$ to bound the first and second expectation term respectively. It follows that $\Pr\left[\gamma_{1,t_{\text{stop}}} = T+1\right] \geq \Delta/4$ as claimed.

□

Recall that $N_{\mathcal{P}} > \frac{\beta_{2,0}}{\alpha_{2,0}}$ and define

$$\texttt{Ev}_2 = \left\{\texttt{tape}_{2,\ell} = 0 \text{ for all } \ell \in [N_{\mathcal{P}}]\right\}.$$

**Lemma C.4.** *Assume* $\texttt{Ev}_1$ *and* $\texttt{Ev}_2$ *hold. Then arm 2 is pulled at most* $N_{\mathcal{P}}$ *times.*

*Proof.* Suppose, for contradiction, that arm 2 is pulled $N_{\mathcal{P}}+1$ times. Let $t$ be the round of the $(N_{\mathcal{P}}+1)$-th pull. Since $\texttt{Ev}_1$ holds, Lemma C.2 implies

$$\gamma_{1,t-1} \geq \gamma_{1,0} - \Delta/4.$$

**Claim C.5.** $\frac{\alpha_{2,t-1}+1}{\alpha_{2,t-1}+\beta_{2,t-1}+1} \leq \gamma_{2,0}$.

*Proof.* Since arm 2 has been pulled $N_{\mathcal{P}}$ times with all rewards equal to 0 (because $\texttt{Ev}_2$ holds), we have

$$\alpha_{2,t-1} = \alpha_{2,0}, \quad \beta_{2,t-1} = \beta_{2,0} + N_{\mathcal{P}}.$$

Therefore,

$$\frac{\alpha_{2,t-1}+1}{\alpha_{2,t-1}+\beta_{2,t-1}+1} = \frac{\alpha_{2,0}+1}{\alpha_{2,0}+\beta_{2,0}+N_{\mathcal{P}}+1}.$$

Let $\alpha = \alpha_{2,0}$ and $\beta = \beta_{2,0}$. By definition of $\gamma_{2,0}$, we have $\gamma_{2,0} = \frac{\alpha}{\alpha+\eta}$. It therefore suffices to show

$$\frac{\alpha+1}{\alpha+\beta+N_{\mathcal{P}}+1} < \frac{\alpha}{\alpha+\beta}.$$

Cross-multiplying and simplifying:

$$(\alpha+1)(\alpha+\beta) < \alpha\,(\alpha+\beta+N_{\mathcal{P}}+1)$$
$$\alpha^2 + \alpha\beta + \alpha + \beta < \alpha^2 + \alpha\beta + \alpha N_{\mathcal{P}} + \alpha$$
$$\beta < \alpha N_{\mathcal{P}},$$

which holds strictly because $N_{\mathcal{P}} > \beta/\alpha$. $\qquad\square$

Since we assumed $\gamma_{1,0} > \gamma_{2,0}$, it follows that

$$\frac{\alpha_{2,t-1}+1}{\alpha_{2,t-1}+\beta_{2,t-1}+1} \le \gamma_{2,0} < \gamma_{1,0} - \Delta/4 \le \gamma_{1,t-1}.$$

By Lemma C.1, arm 2 cannot be played in this round, contradicting the assumption that it is played more than $N_{\mathcal{P}}$ times. $\quad\square$

*Proof of Theorem 4.6(a).* Set $\tau$ such that $\Pr\left[\mu_1 \notin (\tau, 1-\tau)\right] = \Delta/16$. We note that, since $\alpha_{1,0}, \beta_{1,0} > 0$, we have $\gamma_{1,0} \in (0,1)$ which in turn implies $\tau > 0$; we also have $\tau < 1/2$. By Lemma C.3, we have $\Pr\left[\mathtt{Ev}_1\right] \ge \Delta/4$ which by union bound implies

$$\Pr\left[\mu_1 \in (\tau, 1-\tau) \text{ and } \mathtt{Ev}_1\right] \ge \Delta/8.$$

Note that both events depend only on the reward tape for arm 1 and the draw of $\mu_1$ and are independent of $\mu_2$ and reward tape of $\mu_2$.

Now, set $\kappa = \tau/2$. Note that $\tau > 0$ so $\kappa > 0$. Let $\mathtt{Ev}_3$ be the event that $\mu_2 \in (1-\kappa, 1-\kappa/2)$. We note that the probability for $\mathtt{Ev}_3$ is strictly positive because the density function is strictly positive. Since the probability only depends on the prior, we can write it as $\Theta_{\mathcal{P}}(1)$. Condition on $\mathtt{Ev}_3$, the probability of $\mathtt{Ev}_2$ is $\Theta_{\mathcal{P}}(1)$ as well. Therefore, with probability $\Theta_{\mathcal{P}}(1)$ we have $\mathtt{Ev}_3$ and $\mathtt{Ev}_2$. We note that both events depend on the draw of $\mu_2$ and the reward tape of $\mu_2$ and are independent of arm 1.

Since the event $\{\mu_1 \in (\tau, 1-\tau) \text{ and } \mathtt{Ev}_1\}$ is independent of $\{\mathtt{Ev}_2 \text{ and } \mathtt{Ev}_3\}$, it follows that, with probability $\Theta_{\mathcal{P}}(1)$, we have $\mathtt{Ev}_1, \mathtt{Ev}_2, \mathtt{Ev}_3$ and $\mu_1 \in (\tau, 1-\tau)$. Set $c_{\mathcal{P}} = \kappa/2$ (note that $\kappa$ depends only on the prior). In this case, we have $\mu_2 \ge \mu_1 + c_{\mathcal{P}}$ because $\mu_2 \ge 1-\kappa = 1-\tau/2$ (because of $\mathtt{Ev}_3$) while $\mu_1 \le 1-\tau = 1-2\kappa$. We additionally have $\mu_1 \in (c_{\mathcal{P}}, 1-c_{\mathcal{P}})$ because $\mu_1 \in (\tau, 1-\tau)$ and $\mu_2 \in (c_{\mathcal{P}}, 1-c_{\mathcal{P}})$ because $\mu_2 \in (1-\kappa, 1-\kappa/2)$. Additionally, since $\mathtt{Ev}_1, \mathtt{Ev}_2$ hold, by Lemma C.4, arm 2 is pulled at most $N_{\mathcal{P}}$ times, finishing the proof. $\quad\square$

### C.1. Proof of Lemma C.1

*Proof of Lemma C.1.* We will show that any deterministic policy playing arm 2 in the beginning is strictly sub-optimal in maximizing $\mathtt{PU}_\pi$. Since a randomized policy is a distribution over deterministic policies, this implies that even a randomized policy should play this arm with probability 0, which in turn implies the lemma.

For the rest of the proof, we will assume that there exists a posterior optimal (i.e., maximizing $\mathtt{PU}_\pi$) deterministic policy playing arm 2 in round $t$ and use this to obtain a contradiction. Throughout the proof, we use $r_1, r_2$ to denote the rewards of the first and second rounds in the episode. We will often drop the dependence on $t-1$ in the notion $\gamma_{1,t-1}$, writing $\gamma_1$ instead.

We start with the following claim which handles the case where $t$ is the second round of the episode. We will actually prove a stronger statement which uses a weaker assumption than what is stated in the lemma. The stronger statement will be used in the future parts of the proof for handling the case where $t$ is the first round of the episode.

**Claim C.6.** *If $t$ is the second round of the episode and $\gamma_{2,t-1} < \gamma_{1,t-1}$, then arm 2 will not be played in round $t$.*

*Proof.* Since $\gamma_1 > \gamma_2$ and $f(r_1, x)$ is increasing in $x$, the posterior utility if we play arm 1 is at least as high as playing arm 2. If the posterior utility is strictly higher, then the proof is complete. If the two posterior utilities are equal, letting $r_1$ denote the reward of the previous episode,

$$\gamma_2 f(r_1, 1) + (1 - \gamma_2) f(r_1, 0) = \gamma_1 f(r_1, 1) + (1 - \gamma_1) f(r_1, 0).$$

Since $\gamma_1 > \gamma_2$, this implies that $f(r_1, 0) = f(r_1, 1)$. If this is the case however, then skipping leads to a strictly better posterior utility than choosing either arm because it does not incur the cost $c_{\text{sel}}$. □

Given the above lemma, we assume $t$ is the first round of the episode for the rest of the proof. Note that, if arm 2 is chosen in round $t$ then by assumption we have

$$\gamma_{2,t} \leq \frac{\alpha_{1,t-1} + 1}{\alpha_{1,t-1} + \beta_{1,t-1} + 1} \leq \gamma_{1,t-1} = \gamma_{1,t}.$$

Therefore, by Claim C.6, arm 2 will not be chosen in the second round of the episode. Since the only other choices are playing arm 1 and skipping, and the reward in round $t$ is either $0$ or $1$, there are $4$ total cases we need to consider for the posterior optimal policy's behavior in the second round of the episode:

- Case I: always skip.

- Case II: always play arm 1.

- Case III: if $r_1 = 0$, play arm 1; if $r_1 = 1$ skip.

- Case IV: if $r_1 = 1$, play arm 1; if $r_1 = 0$ skip.

Note that we have already assumed the posterior optimal policy is playing arm 2 in the first round. We separately consider each of the 4 cases and show a contradiction in each one. In all 4 cases, we use $\gamma_1^+$ to denote $\frac{\alpha_{1,t-1}+1}{\alpha_{1,t-1}+\beta_{1,t-1}+1}$.

**Case I.** The policy's posterior utility is $-c_{\text{sel}} + \gamma_2 f(1, 0) + (1 - \gamma_2) f(0, 1)$. This implies that $f(1, 0)$ must be strictly higher than $f(0, 0)$. If it is not, then we can just skip in the first round and get $f(1, 0)$ which is strictly higher posterior utility because it does not have the $-c_{\text{sel}}$ term. Given this, if we play arm 1 instead of arm 2, we would have a an increase of $(\gamma_1 - \gamma_2)(f(1, 0) - f(0, 1)) > 0$ in the posterior utility which is non-negative.

**Case II.** If we play arm 1 first and then arm 2, this would give us the same utility; we pay $2c_{\text{sel}}$ for the cost in both cases and we get $f(1, 1), f(1, 0) = f(0, 1)$, and $f(0, 0)$ with probabilities $\gamma_1 \gamma_2$, $\gamma_1(1 - \gamma_2) + \gamma_2(1 - \gamma_1)$, and $(1 - \gamma_1)(1 - \gamma_2)$ respectively. Now, consider a different policy that plays arm 1 and if it sees 1, it plays 1 again; otherwise it plays 2. The posterior utility of this policy is higher by $\gamma_1(\gamma_1^+ - \gamma_2)(f(1, 1) - f(1, 0))$, which is strictly higher than 0 unless $f(1, 1) = f(1, 0)$. If $f(1, 1) = f(1, 0)$ however, then the policy that plays arm 1 and then skips if it sees 1, playing arm 2 otherwise, would have an posterior utility which is higher by $\gamma_1 c_{\text{sel}}$.

**Case III.** The posterior utility equals

$$-c_{\text{sel}} + \gamma_2 f(1, 0) + (1 - \gamma_2)(-c_{\text{sel}} + \gamma_1 f(0, 1)) = -c_{\text{sel}} + (\gamma_2 + \gamma_1 - \gamma_1 \gamma_2) f(0, 1) - (1 - \gamma_2) c_{\text{sel}}.$$

If we were to play arm 1, skipping if we get 1 and playing arm 2 if we get 0, then we would get the posterior utility

$$-c_{\text{sel}} + \gamma_1 f(1, 0) + (1 - \gamma_1)(-c_{\text{sel}} + \gamma_2 f(0, 1)) = -c_{\text{sel}} + (\gamma_1 + \gamma_2 - \gamma_1 \gamma_2) f(1, 0) - (1 - \gamma_1) c_{\text{sel}}.$$

Since $\gamma_1 > \gamma_2$, we have $-(1 - \gamma_1) > -(1 - \gamma_2)$, which means the alternative policy has strictly higher utility.

**Case IV.** The posterior utility equals

$$-c_{\text{sel}} + \gamma_2(-c_{\text{sel}} + \gamma_1 f(1,1) + (1-\gamma_1)f(1,0))$$

The alternative policy of playing 1 first, then playing it again if we get 1 has utility

$$-c_{\text{sel}} + \gamma_1(-c_{\text{sel}} + \gamma_1^+ f(1,1) + (1-\gamma_1^+)f(1,0))$$

Since $\gamma_1^+ \geq \gamma_1$ and $f(1,1) \geq f(1,0)$, this is at least

$$-c_{\text{sel}} + \gamma_1(-c_{\text{sel}} + \gamma_1 f(1,1) + (1-\gamma_1)f(1,0))$$

The posterior utility of the alternative minus the posterior utility of the original equals

$$(\gamma_1 - \gamma_2)(-c_{\text{sel}} + \gamma_1 f(1,1) + (1-\gamma_1)f(1,0)).$$

If $(-c_{\text{sel}} + \gamma_1 f(1,1) + (1-\gamma_1)f(1,0)) > 0$, we are done. Otherwise, the policy's posterior utility is at most $-c_{\text{sel}}$, which means it loses to the policy which skips both rounds. □

## D. Proof of Theorem 4.1: Symmetric Functions

In this section, we prove Theorem 4.1. Observe that symmetric functions, since the input is binary, the value of $f(x_1, \ldots, x_m)$ depends only on $\sum_i x_i$. Therefore, throughout the section we abuse notation and right $f(x)$ for $x \in \{0, 1, \ldots, m\}$ to denote $f(x_1, \ldots, x_m)$ where exactly $x$ values in $(x_1, \ldots, x_m)$ are equal to 1.

We proceed with a proof. For any $i \geq 1$, define

$$\rho_i = \prod_{\ell=0}^{i} \frac{\alpha_{1,t-1}}{\alpha_{1,t-1} + \beta_{1,t-1} + \ell} \tag{6}$$

We will assume that, for any $i, j, m$ satisfying $j \leq i \leq m$ we have

$$\rho_{m-i}(f(m-i+j) - f(j)) > (m-i)c_{\text{sel}} \text{ or } f(m-i+j) = f(j). \tag{7}$$

As we will see, with a suitable choice of $c_{\max}$ in Theorem 4.1, Equation (7) follows from the assumption $c_{\text{sel}} \leq c_{\max}$.

To better understand Equation (7), assume that the agent plays for $i$ rounds out of the $m$ possible rounds without using arm 1 and observes $j$ draws equal to 1. At this point, If the agent skips for all $m-i$ rounds, it will always obtain $f(j+m-i)$ for its score. If the agent chooses arm 1 for all of the remaining $m-i$ rounds, its score will be between $f(j)$ and $f(j+m-i)$. Moreover, the probability that the agent obtains $f(j+m-i)$ is lower bounded by $\rho_{m-i}$. The assumption ensures that, if $f(j+m-i)$ is strictly greater than $f(j)$, then the policy of playing arm 1 for all of the remaining rounds strictly beats the policy of skipping for all of the remaining rounds. This is because, the expected increase in score outweighs the cost of playing $m-i$ additional rounds.

We note that, instanitating $(i,j) = (1,1)$ and $(i,j) = (1,0)$, we obtain the following two inequalities as a consequence of the assumption

$$\rho_{m-1} \cdot (f(m) - f(1)) > (m-1)c_{\text{sel}} \text{ or } f(m) = f(1). \tag{8}$$

and

$$\rho_{m-1} \cdot (f(m-1) - f(0)) > (m-1)c_{\text{sel}} \text{ or } f(m-1) = f(0). \tag{9}$$

We further observe that the assumption is inductive in the sense that if it holds at the beginning of the episode and we play arm 2, then looking at the rest of the episode as an episode of length $m-1$, then the assumption will still hold. Formally, the following lemma holds

**Lemma D.1.** *Fix $m \geq 1$ and $f$. Assume that Equation (7) holds for all $i, j$ sastifying $j \leq i \leq m$. For some values $i, j$ such that $j \leq i$ define $g : \{0, \ldots, m-i\} \to \mathbb{R}$ as $f(\ell + j)$. The function $g$ satisfies Equation (7) for all $i', j'$ satisfying $j' \leq i' \leq m-i$. Formally, for any $i', j'$ such that $j' \leq i' \leq m-i$ we have*

$$\rho_{m-i-i'}(g(m-i-i'+j') - g(j')) > (m-i-i')c_{\text{sel}} \text{ or } g(m-i-i'+j') = g(j')$$

*Proof.* By definition of $g$, we need to show that

$$\rho_{m-i-i'}(f(m-i-i'+j'+j) - f(j'+j)) > (m-i-i')c_{\text{sel}} \text{ or } f(m-i-i'+j'+j) = f(j'+j)$$

This is the same as Equation (7) for $(i+i', j+j')$. Note that we have $j + j' \leq i + i'$ because $j \leq i$ and $j' \leq i'$. We also have $i' + i \leq m$ because $i' \leq m - i$. $\qquad\square$

We next focus on a special case which can effectively be thought of as $f(x_1, \ldots, x_m) = \min\{x_1, \ldots, x_m\}$.

**Lemma D.2.** *Assume that $f : \{0,1\}^m \to \mathbb{R}$ is symmetric, coordinate-wise increasing and $f(m-1) = f(0)$. Assume further that $\frac{\alpha_{2,t-1}+m-1}{\alpha_{2,t-1}+\beta_{2,t-1}+m-1} < \gamma_{1,t-1}$. Any deterministic posterior optimal policy has one of the following two forms.*

- *Skip all rounds.*

- *Start by playing arm $1$ and continue doing so while the observed rewards are $1$; skip all remaining rounds upon observing a reward of $0$.*

*Proof.* The case of $m = 1$ follows from Claim C.6. We therefore focus on $m \geq 2$. We prove the claim using induction, assuming that it holds for $m - 1$. Throughout, we say a policy chooses *abort* at some round if plays skip for that round and all future rounds of the episode.

We assume without loss of generality that $f(0) = 0$. We first observe that if a deterministic posterior optimal policy ever sees the reward $0$ then it must skip all future rounds. This holds because, $f(m-1) = f(0)$ which means future rewards will not change score. Since the cost is strictly positive, the policy should skip.

We next claim that if a deterministic posterior optimal policy does not abort from the beginning (i.e., it playes either arm $1$ or $2$ in round $1$) then it must keep playing either arm $1$ or $2$ as long as it observes the reward $1$. To see why, assume for contradiction that it aborts on round $i$ if the rewards from rounds $1$ to $i-1$ are all $1$. Since we have already shown that the policy would abort if it sees a reward of $0$ during rounds $1$ to $i-1$, then it follows that the policy *always* aborts after round $i$. This means however that its score (i.e., $f(r_t, \ldots, t_{t+m-1})$) is always $0$. As such, its utility is always negative since it needs to pay a cost of $c_{\text{sel}}$ for playing the first round. It follows that this policy is strictly worse than skipping all rounds.

Given the above claim, if a deterministic policy is posterior optimal and it does not skip all rounds, then there is a sequence $\texttt{seq}$ of $m$ values in $\{1, 2\}$ such that, as long as the previously observed rewards are $1$, in round $i$ the policy should pull the arm specified by the $i$-th value in $\texttt{seq}$ and, upon observing a reward of $0$, the policy should skip all future rounds.

Define $g^+ : \{0, \ldots, m-1\} \to \mathbb{R}$ as $g^+(\ell) = f(\ell + 1)$. If the posterior optimal policy starts by playing arm 1, we claim that it will keep playing arm 1 throughout the rest of the episode as long as the rewards equal to $1$. This follows from the induction hypothesis. Specifically, after playing the first arm $1$ if the policy observes the reward $1$, then the remainder of the episode can be thought of as instance of the same problem with episode length $m - 1$ and the function $g^+$ instead of $f$. Note that we have $g^+(m-2) = f(m-1) = f(1) = g(0)$. Moreover,

$$\frac{\alpha_{2,t}+m-2}{\alpha_{2,t}+\beta_{2,t}+m-2} = \frac{\alpha_{2,t-1}+m-2}{\alpha_{2,t-1}+\beta_{2,t-1}+m-2} < \frac{\alpha_{2,t-1}+m-1}{\alpha_{2,t-1}+\beta_{2,t-1}+m-1} < \gamma_{1,t-1} < \gamma_{1,t}.$$

Therefore, the preconditions of the induction hypothesis hold and the policy should either play only arm $1$ or skip all future rounds. As mentioned however, since the policy has already played once $1$, it should not skip, finishing the proof.

Therefore, if the policy plays arm $1$ in the first round, the lemma is proved. Similarly, if the policy skips the lemma is proved because, since skipping a round leads to reward $0$ for that round, the policy should skip for future rounds. To finish the proof, it suffices to show that any deterministic posterior optimal policy will not start by playing arm $2$.

Assume (for contradiction) that a posterior optimal policy starts by playing arm $2$. We claim that the associated sequence for this policy should be $[2, 1, \ldots, 1]$; i.e., the policy starts with arm $2$ and uses arm $1$ for all subsequent rounds. This again follows from the induction hypothesis. As before, we have an episode of length $m - 1$ with $g^+$. The preconditions also hold because

$$\frac{\alpha_{2,t}+m-2}{\alpha_{2,t}+\beta_{2,t}+m-2} = \frac{\alpha_{2,t-1}+m-1}{\alpha_{2,t-1}+\beta_{2,t-1}+m-1} < \gamma_{1,t-1} = \gamma_{1,t}.$$

It follows that the associated sequence is $[2, 1, \ldots, 1]$.

We proceed by defining some notation. For any values $x_1, \ldots, x_m \in \{0, 1\}$, let $u(x_1, \ldots, x_m)$ denote the expected utility if in round $i$ we play an arm that, conditioned on the previous rounds, yields the reward 1 with probability $x_i$. We can define $u(x_1, \ldots, x_m)$ recursively as

$$u(x_m) = -c_{\text{sel}} + x_m f(m), \quad u(x_i, \ldots, x_m) = -c_{\text{sel}} + x_i u(x_{i+1}, \ldots, x_m).$$

Define $\gamma_{1,t-1}^{+,(i)}$ for $i \geq 1$ as the posterior mean of arm 1 if we see $i$ rewards equal to 1; i.e., $\gamma_{1,t-1}^{+,(i)} = \frac{\alpha_{1,t-1}+i}{\alpha_{1,t-1}+\beta_{1,t-1}+i}$. For the remainder of the proof, we drop the dependence on $t-1$ in $\gamma_{1,t-1}$. Set $(x_1, \ldots, x_m) = (\gamma_2, \gamma_1, \gamma_1^{+,(1)}, \ldots, \gamma_1^{+,(m-2)})$.

We will show that either $u(x_1, \ldots, x_m) < 0$ or, $u(\gamma_1, \gamma_1^{+,(1)}, \ldots, \gamma_1^{+,(m-1)}) > u(x_1, \ldots, x_m)$. We claim that this finishes the proof. In the first case, the posterior utility for the policy starting with arm 2 is negative and as such it is not posterior optimal because the posterior utility of skipping all rounds is 0. In the second case, the policy playing arm 1 in all rounds will beat the policy of starting with arm 2.

It remains to show that either $u(x_1, \ldots, x_m) < 0$ or, $u(\gamma_1, \gamma_1^{+,(1)}, \ldots, \gamma_1^{+,(m-1)}) > u(x_1, \ldots, x_m)$. Assume that $u(x_1, \ldots, x_m) \geq 0$; we need to show that $u(\gamma_1, \gamma_1^{+,(1)}, \ldots, \gamma_1^{+,(m-1)}) > u(x_1, \ldots, x_m)$. We first claim that for any $i > 1$, $u(x_i, \ldots, x_m) > 0$. If this does not hold, then by definition of $u(.)$, we have $u(x_{i-1}, \ldots, x_m) = -c_{\text{sel}} + x_{i-1} u(x_i, \ldots, x_m) < 0$. Repeating this reasoning for smaller values of $i$, we obtain that $u(x_1, \ldots, x_m) < 0$, which is a contradiction.

We next show via (backwards) induction that if $x_1', \ldots, x_m'$ satisfy $x_i' > x_i$ for all $i$, then for any $i$ we have $u(x_i', x_{i+1}', \ldots, x_m') > u(x_i, x_{i+1}, \ldots, x_m)$. If this is shown then setting $i = 1$ and using the fact that $\gamma_1 > \gamma_2$ and $\gamma_1^{+,(i)} > \gamma_1^{+,(i-1)}$ finishes the proof. We start by verifying the base case of $i = m$. If $f(m) \leq 0$, then $u(x_m) < 0$ which contradicts the previous claim. Therefore, $f(m) > 0$ which implies $u(x_m') > u(x_m)$. Assuming the claim holds for $i + 1$, we have

$$
\begin{aligned}
u(x_i', \ldots, x_m') &= -c_{\text{sel}} + x_i' u(x_{i+1}', \ldots, x_m') && \textit{(Definition of } u(.)) \\
&> -c_{\text{sel}} + x_i' u(x_{i+1}, \ldots, x_m) && \textit{(Since } x_i' > x_i \geq 0 \textit{ and induction hypothesis)} \\
&\geq -c_{\text{sel}} + x_i u(x_{i+1}, \ldots, x_m) && \textit{(Since } u(.) > 0) \\
&= u(x_i, \ldots, x_m), && \textit{(Definition of } u(.))
\end{aligned}
$$

finishing the proof.

$\square$

**Lemma D.3.** *Assume that $f(\cdot, \cdot, \ldots, \cdot) : \{0, 1\}^m \to \mathbb{R}$ is symmetric, coordinate-wise increasing.* $\frac{\alpha_{2,t-1}+m-1}{\alpha_{2,t-1}+\beta_{2,t-1}+m-1} < \gamma_{1,t-1}$. *Assume further that Equation (7) holds. Arm 2 will not be selected in round $t$.*

*Proof of Lemma D.3.* As before, we will show that any deterministic policy playing arm 2 is strictly inferior to some alternative deterministic policy that does not play arm 2. This implies the lemma as any randomized policy should put weight 0 on the deterministic policy playing arm 2.

We prove by induction on $m$. The case where $m = 2$ is already done by Lemma C.1. Assuming the claim holds for $1, \ldots, m - 1$, we will show it holds for $m$ as well. We assume without loss of generality that $t$ is the first round of the episode. If this does not hold, then the lemma follows from the induction hypothesis. Concretely, the remainder of the episode is an episode of length $m' < m$. Assume $m' = m - i$ and that we have seen $j$ values of 1 in the first $i$ episodes. Define $g : \{0, 1, \ldots, m'\}$ as $g(\ell) = f(\ell + j)$. By Lemma D.1, the function $g$ satisfies Equation (7) for $j' \leq i' \leq m'$. We additionally have

$$\frac{\alpha_{2,t-1}+m-1}{\alpha_{2,t-1}+\beta_{2,t-1}+m-1} \leq \frac{\alpha_{2,t-1}+m'-1}{\alpha_{2,t-1}+\beta_{2,t-1}+m'-1} < \gamma_{1,t-1}.$$

Therefore, the preconditions of the induction hypothesis hold and arm 2 will not be played. Therefore, throughout the rest of the proof we assume that $t$ is not the first round of the episode.

Assume for the sake of contradiction that we start the episode by playing arm 2. Afterwards, the posterior optimal policy cannot play arm 2 in the next round because of induction hypothesis. Formally, let $r_1$ denote the reward of the first round in the episode, define $g^+, g^- : \{0, 1, \ldots, m-1\}$ as $g^+(\ell) = f(\ell+1)$ and $g^-(\ell) = f(\ell)$. the remainder of the episode can be seen as an episode of length $m-1$ where $f$ is replaced with either $g^+$ or $g^-$ depending on whether $r_1 = 1$ or $r_1 = 0$. The preconditions of the induction hypothesis holds; we have $\gamma_{1,t} > \frac{\alpha_{2,t}+m-2}{\alpha_{2,t}+\beta_{2,t}+m-2}$ because $\alpha_{2,t} \leq \alpha_{2,t-1}+1$, $\beta_{2,t} \geq \beta_{2,t-1}$, $\gamma_{1,t} = \gamma_{1,t-1}$ and $\frac{\alpha_{2,t-1}+m-1}{\alpha_{2,t-1}+\beta_{2,t-1}+m-1} < \gamma_{1,t-1}$ by assumption. Equation (7) for either $g^+$ or $g^-$ would also hold by Lemma D.1.

There are therefore 4 possibilities for what the policy will do in the second round:

- Case I: always skip.

- Case II: always play arm 1.

- Case III: if $r_1 = 0$, play arm 1; if $r_1 = 1$ skip.

- Case IV: if $r_1 = 1$, play arm 1; if $r_1 = 0$ skip.

As before, we say a policy chooses *abort* at some round if plays skip for that round and all future rounds of the episode. We first claim that skip actions can be pushed to the end of the episode without loss of optimality, which in turn means that if the posterior optimal policy does skip the second round then it chooses to abort. Since the reward function is symmetric and skipping leads to a deterministic reward of 0, if a policy skips some rounds and then chooses either arm 1 or 2 at some later point, its posterior utility is equal to an alternative policy that plays that arm first and then skips the same number of rounds. In other words, we can assume that the posterior optimal policy de-prioritizes skip moves: it only uses skip if it intends to play skip for the remainder of the episode.

**Case I.** Since the policy skips the remainder of the episode, it score is equal to $f(r_1) \in \{f(1), f(0)\}$. If $f(1) = f(0)$, then the policy that skips first round would do better. If $f(1) > f(0)$, then playing arm 1 in the first round would lead to a strictly higher utility. In both cases, playing arm 2 is strictly sub-optimal and the proof is complete.

**Case II.** Instead of playing arm 2 and then 1, let us assume we play arm 1 and then 2. We play the rest of the rounds the same way we would have normally; specifically, after observing the rewards $(r_1, \ldots, r_\ell)$ for some $\ell \in [3, m]$, we choose the same arm as the posterior optimal policy would make after observing rewards $(r_2, r_1, r_3, \ldots, r_\ell)$. This gives the same posterior utility as the posterior optimal policy because we have merely switched the reward of the first two rounds. We note that the probability of observing the rewards $(r_1, r_2, \ldots, r_m)$ under the new policy is the same as the probability of observing the rewards $(r_2, r_1, r_3, \ldots, r_m)$ under the old policy.

Since the original policy was posterior optimal, the current policy should be posterior optimal as well. We observe however that if $r_1 = 1$, then playing arm 2 is not posterior optimal. This is because, if we look at the rest of the $m-1$ rounds, we effectively have an episode of length $m-1$ with $g^+ : \{0, \ldots, m-1\}$ instead of $f$ where $g^+(\ell) = f(\ell+1)$. By induction, arm 2 will not be played. Note that the preconditions of the induction hold. To show $\frac{\alpha_{2,t}+m-2}{\alpha_{2,t}+\beta_{2,t}+m-2} < \gamma_{1,t}$, observe that since we played arm 1 in the first round we have $(\alpha_{2,t}, \beta_{2,t}) = (\alpha_{2,t-1}, \beta_{2,t-1})$ and since $r_1 = 1$ we have $\gamma_{1,t} > \gamma_{1,t-1}$. Since $\frac{\alpha_{2,t}+m-2}{\alpha_{2,t}+\beta_{2,t}+m-2} \leq \frac{\alpha_{2,t}+m-1}{\alpha_{2,t}+\beta_{2,t}+m-1}$ and we we already have $\frac{\alpha_{2,t-1}+m-1}{\alpha_{2,t-1}+\beta_{2,t-1}+m-1} < \gamma_{1,t-1}$, the condition follows. Equation (7) for $g^+(.)$ holds by Lemma D.1. Specifically, defining $m' = m-1$ and $\rho'_i = \prod_{\ell=0}^{i} \frac{\alpha_{1,t}}{\alpha_{1,t}+\beta_{1,t}+\ell}$ similar to Equation (6), we have $\rho'_i \geq \rho_i$. Additionally, $\rho_{m'-i}$ and $g^+(.)$ satisfy Equation (7) for $j \leq i \leq m'$ by Lemma D.1. Since $\rho'_{m'-i} \geq \rho_{m'-i}$, it follows that $\rho'_{m'-i}$ and $g^+(.)$ also satisfy Equation (7) and the preconditions of the induction hold. Therefore, by induction assumption, it is strictly better to choose an option other than arm 2 in the second round if $r_1 = 1$. The event $r_1 = 1$ happens with strictly positive probability because $\gamma_{1,t-1} > \gamma_{2,t-1} \geq 0$. It follows that playing arm 1 and then arm 2 in the first two rounds cannot be posterior optimal, finishing the proof.

**Case III.** We first consider the case $f(m) > f(1)$. For small enough cost it is worth it to play some arm rather than skip. Specifically, the probability that we see $m-1$ draws of 1 if we keep playing is at least $\rho_{m-1}$; so in expectation we get an extra $\rho_{m-1}(f(m) - f(1))$ in the score which outweights the cost by Equation (8).

If $f(m) = f(1)$, then if $r_1 = 0$ and we play arm 1, we would also abort if we see $r_2 = 1$. Therefore, the utility is going to be

$$-c_{\text{sel}} + \gamma_2 f(1) + (1 - \gamma_2)(-c_{\text{sel}} + \gamma_1 f(1) + (1 - \gamma_1)X),$$

for some value $X$. If we play arm 1 and then 2 if $r_1 = 0$ however,

$$-c_{\text{sel}} + \gamma_1 f(1) + (1 - \gamma_1)(-c_{\text{sel}} + \gamma_2 f(1) + (1 - \gamma_2)Y).$$

for some value $Y$. We observer however that $X = Y$ because, having observed a reward of $0$ for both arm 1 and 2, the order in which the rewards are seen does not affect the posterior update and, as such, the posterior utility conditioned on the rewards. Letting $\text{PU}_{\text{alt}}$ and $\text{PU}_{\text{opt}}$ denote the posterior utility of the alternative policy and the posterior optimal policy respectively, we conclude that

$$\text{PU}_{\text{alt}} - \text{PU}_{\text{opt}} = (\gamma_1 - \gamma_2)(f(1) + c_{\text{sel}} - f(1)) = (\gamma_1 - \gamma_2)c_{\text{sel}}$$

Therefore $\text{PU}_{\text{alt}} > \text{PU}_{\text{opt}}$.

**Case IV.** If $f(m-1) > f(0)$, then the posterior optimal policy should not skip with $r_1 = 0$ which is a contradiction. Specifically, the probability that we get $m - 1$ draws of 1 next is at least $\rho_{m-1}$. If $f(m-1) > f(0)$, then by Equation (9) we have $\rho_{m-1}(f(m-1) - f(0)) > (m-1)c_{\text{sel}}$ which means it is strictly better to play $m - 1$ draws of arm 1 instead of skipping all remaining rounds. Therefore, if $f(m-1) > f(0)$ then the current case will not occur which is a contradiction. The case where $f(m-1) = f(0)$ follows from Lemma D.2.

$\square$

We next prove a lemma that provides a conditional lower bound on $\rho_i$ which is independent of $t$. This allows to ensure Equation (7) as long as the condition holds.

**Lemma D.4.** *For any $t \geq 1$, if $\gamma_{1,t-1} \geq \gamma_{1,0} - \Delta/4$ then*

$$\rho_i \geq \prod_{\ell=0}^{i} \frac{\gamma_{1,0} - \Delta/4}{1 + \frac{\ell}{\alpha_{1,0} + \beta_{1,0}}}$$

*Proof.* Note that

$$\frac{\alpha_{1,t-1}}{\alpha_{1,t-1} + \beta_{1,t-1} + \ell} = \frac{(\alpha_{1,t-1})/(\alpha_{1,t-1} + \beta_{1,t-1})}{(\alpha_{1,t-1} + \beta_{1,t-1} + \ell)/(\alpha_{1,t-1} + \beta_{1,t-1})} = \frac{\gamma_{1,t-1}}{1 + \frac{\ell}{\alpha_{1,t-1} + \beta_{1,t-1}}}$$

Note however that $\alpha_{1,t-1} + \beta_{1,t-1} \geq \alpha_{1,0} + \beta_{1,0}$, which implies

$$\frac{1}{1 + \frac{\ell}{\alpha_{1,t-1} + \beta_{1,t-1}}} \geq \frac{1}{1 + \frac{\ell}{\alpha_{1,0} + \beta_{1,0}}}.$$

Since $\gamma_{1,t-1} \geq \gamma_{1,0} - \Delta/4$, we have

$$\frac{\alpha_{1,t-1}}{\alpha_{1,t-1} + \beta_{1,t-1} + \ell} \geq \frac{\gamma_{1,0} - \Delta/4}{1 + \frac{\ell}{\alpha_{1,0} + \beta_{1,0}}}.$$

Taking a product over $\ell \in [0, i]$ finishes the proof. $\square$

Recall that $N_{\mathcal{P}} = \left\lfloor (m-1)\frac{\beta_{2,0}}{\alpha_{2,0}} \right\rfloor + 1$ and define

$$\text{Ev}_2 = \left\{ \texttt{tape}_{2,\ell} = 0 \text{ for all } \ell \in [N_{\mathcal{P}}] \right\}. \tag{10}$$

Define the arm1-stability event $\text{Ev}_1$ as in Equation (5).

**Lemma D.5.** *Assume* $\mathrm{Ev}_1$ *and* $\mathrm{Ev}_2$ *hold. Assume further that for any* $i, j, m$ *such that* $j \leq i \leq m$ *we have*

$$\left( \prod_{\ell=0}^{m-i} \frac{\gamma_{1,0} - \Delta/4}{1 + \frac{\ell}{\alpha_{1,0} + \beta_{1,0}}} \right) (f(m - i + j) - f(j)) > (m - i)c_{\text{sel}} \text{ or } f(m - i + j) = f(j). \tag{11}$$

*Then arm* 2 *is pulled at most* $N_{\mathcal{P}}$ *times.*

*Proof.* Suppose, for contradiction, that arm 2 is pulled $N_{\mathcal{P}} + 1$ times. Let $t$ be the round of the $(N_{\mathcal{P}} + 1)$-st pull. Since $\mathrm{Ev}_1$ holds, Lemma C.2 implies

$$\gamma_{1,t-1} \geq \gamma_{1,0} - \Delta/4.$$

Additinonally, Lemma D.4 and Equation (11) imply Equation (7).

**Claim D.6.** $\frac{\alpha_{2,t-1} + m - 1}{\alpha_{2,t-1} + \beta_{2,t-1} + m - 1} \leq \gamma_{2,0}.$

*Proof.* Since arm 2 has been pulled $N_{\mathcal{P}}$ times with all rewards equal to 0 (because $\mathrm{Ev}_2$ holds), we have

$$\alpha_{2,t-1} = \alpha_{2,0}, \quad \beta_{2,t-1} = \beta_{2,0} + N_{\mathcal{P}}.$$

Therefore,

$$\frac{\alpha_{2,t-1} + m - 1}{\alpha_{2,t-1} + \beta_{2,t-1} + m - 1} = \frac{\alpha_{2,0} + m - 1}{\alpha_{2,0} + \beta_{2,0} + N_{\mathcal{P}} + m - 1}.$$

Let $\alpha = \alpha_{2,0}$ and $\beta = \beta_{2,0}$. It suffices to show

$$\frac{\alpha + m - 1}{\alpha + \beta + N_{\mathcal{P}} + m - 1} < \frac{\alpha}{\alpha + \beta}.$$

Cross-multiplying and simplifying:

$$(\alpha + m - 1)(\alpha + \beta) < \alpha (\alpha + \beta + N_{\mathcal{P}} + m - 1)$$
$$\alpha(\alpha + \beta) + (m - 1)(\alpha + \beta) < \alpha(\alpha + \beta) + \alpha(N_{\mathcal{P}} + m - 1)$$
$$(m - 1)\beta < N_{\mathcal{P}}\alpha$$

which holds strictly because $N_{\mathcal{P}} > (m - 1)\beta/\alpha$. $\square$

Since $\gamma_{1,0} - \gamma_{2,0} = \Delta > 0$, it follows that

$$\frac{\alpha_{2,t-1} + m - 1}{\alpha_{2,t-1} + \beta_{2,t-1} + m - 1} \leq \gamma_{2,0} < \gamma_{1,0} - \Delta/4 \leq \gamma_{1,t-1}.$$

By Lemma D.3, arm 2 cannot be played in this round, contradicting the assumption that it is played more than $N_{\mathcal{P}}$ times. $\square$

### D.1. Proof of Theorem 4.1

*Proof of Theorem 4.1.* Define the constant $c_{\max}$ to be the minimum of the following quantity over all $i, j$ satisfying $j \leq i \leq m$:

$$\begin{cases} \frac{1}{m-i} \left( \prod_{\ell=0}^{m-i} \frac{\gamma_{1,0} - \Delta/4}{1 + \frac{\ell}{\alpha_{1,0} + \beta_{1,0}}} \right) (f(m - i + j) - f(j)) & \text{if} \quad f(m - i + j) > f(j) \\ 1 & \text{otherwise.} \end{cases}$$

Note that, since $f(m - i + j) = f(j)$ for $i = m$, the above constant is strictly positive. Additionally, it depends only on the prior and $f(\cdot)$ as required by the theorem statement. If $c_{\text{sel}} < c_{\max}$, then Equation (11) holds for all $i, j$ such that $j \leq i \leq m$.

The remainder of the proof is similar to Theorem 4.6. As before, we have $\Pr[\text{Ev}_1] \geq \Delta/4$ by Lemma C.3. As before, define $\tau$ such that $\Pr[\mu_1 \notin (\tau, 1-\tau)] = \Delta/16$ and note that $\tau > 0$ since $\alpha_{1,0}, \beta_{1,0} > 0$. Taking union bound, we conclude that

$$\Pr[\text{Ev}_1 \text{ and } \mu_1 \in (\tau, 1-\tau)] \geq \Delta/8.$$

Define $\text{Ev}_3$ as $\mu_2 \in (1-\kappa, 1-\kappa/2)$ where $\kappa = \frac{\tau}{2}$. The event holds with probability $\Theta_{\mathcal{P}}(1) > 0$ as before since the density function for arm 2 is strictly positive. Conditioned on $\text{Ev}_3$, we have $\text{Ev}_2$ (see Equation (10)) with probability at least $(\kappa/2)^{N_{\mathcal{P}}}$ which is $(\Theta_{\mathcal{P}}(1))^m$. If all of these hold, then $\mu_2 \geq \mu_1 + c_{\mathcal{P}}$ and FAIL happens. Concretely, we have $\mu_2 \geq 1 - \kappa = 1 - \tau + \kappa \geq \mu_1 + \kappa$. Setting $c_{\mathcal{P}} = \kappa/2$, we have $\mu_2 \geq \mu_1 + c_{\mathcal{P}}$. We also have $\mu_1 \in (c_{\mathcal{P}}, 1 - c_{\mathcal{P}})$ because $\mu_1 \in (\tau, 1-\tau)$ and $\mu_2 \in (c_{\mathcal{P}}, 1 - c_{\mathcal{P}})$ because $\mu_2 \in (1-\kappa, 1-\kappa/2)$. Finally, since $\text{Ev}_2$ and $\text{Ev}_1$ hold and so does Equation (11), by Lemma D.5 we conclude that arm 2 is not played more than $N_{\mathcal{P}}$ times. Since arms 1 and 2 have independent priors and reward tapes, the probability of all of the mentioned events holding is at least $\Theta_{\mathcal{P}}(1)^m$ as required and the proof is complete. $\qquad\square$

## E. Proof of Theorem 4.6(b): function $f$ is $\max$ or $\min$

In this section, we consider the special cases of the $\min$ and $\max$ functions and show that one can obtain Property (P1) without an assumption on $c_{\text{sel}}$. In addition to removing this assumption, we greatly simplify the proof of Theorem 4.1. The simplification is achieved because for these special cases, we can obtain Lemma E.1 below which is a simpler version of Lemma D.3 and, similar to Lemma C.1, does not require an assumption on $c_{\text{sel}}$. This allows us to directly use the proof structure of Theorem 4.6 without accounting for the extra complexities in the proof of Theorem 4.1.

To obtain Lemma E.1, we use the fact that for both $\min$ and $\max$, seeing either a reward of $0$ or a reward of $1$ causes future rounds to be irrelevant. As such, upon observing a reward of $0$ for $\min$, or observing a reward of $1$ for $\max$, the posterior optimal strategy would skip all remaining rounds. Compared to the proof of Lemma D.3 which considered 4 cases in induction, this already simplifies our proof as we only need to consider 2 cases. Additionally, the extra structure greatly reduces the number of policies considered; e.g., for $\min$, a posterior optimal policy must follow a fixed sequence of arms as long as it sees the reward $1$ and must skip all remaining rounds upon seeing the reward $0$.

**Lemma E.1.** *Assume that $f(\cdot, \cdot, \ldots, \cdot) : \{0,1\}^m \to \mathbb{R}$ is symmetric, coordinate-wise increasing, and either $f(1) = f(m)$ or $f(0) = f(m-1)$. Assume $\frac{\alpha_{2,t-1}+m-1}{\alpha_{2,t-1}+\beta_{2,t-1}+m-1} < \gamma_{1,t-1}$. Arm 2 will not be selected in round $t$.*

*Proof.* The $f(0) = f(m-1)$ case follows from Lemma D.2. We therefore focus on the $f(1) = f(m)$ case.

We prove by induction on $m$. For $m = 2$, the claim follows from Lemma C.1. Assume the claim holds for $1, \ldots, m-1$. We show it holds for $m$ as well. As before, we can assume w.l.o.g that $t$ is the first round of the episode. If this is not the case, then the lemma holds by induction hypothesis.

Assume for contradiction that the posterior optimal policy starts by playing arm 2. If $r_1 = 1$, then the policy should clearly skip all remaining rounds as $f(1) = f(m)$. If $r_1 = 0$, then invoking the induction hypothesis for the remainder episode of length $m$, we obtain that the policy should choose either skip or arm 1. We consider the two cases separately.

**Case I.** We assume that the policy skips for $r_1 = 0$. As in the proof of Lemma D.3 we assume w.l.o.g that the policy skips all future rounds since $f$ is symmetric and a policy can prioritize non-skip actions without reducing its posterior utility. Therefore, in this case we always have $r_2 = 0$ which implies the posterior utility equals

$$-c_{\text{sel}} + \gamma_2 f(1).$$

However, if we were to choose arm 1 instead of 2 in the first round, we would get the posterior utility $-c_{\text{sel}} + \gamma_1 f(1)$. Since $f(1) = f(m) > f(1)$, the alternative policy has a strictly higher posterior utility, contradicting the initial optimality assumption.

**Case II.** We assume that the policy plays arm 1 for $r_1 = 0$. If $r_2 = 1$, then the policy should once again skip all remaining rounds. Let $X$ denote the optimal posterior policy for the $m-2$ episode with score function $g(.)$ defined as

$$f'(0) = 0, f'(1) = f'(2) = \ldots f'(m-2) = f(m)$$

and arm parameters

$$(\alpha_1', \beta_1') = (\alpha_1, \beta_1 + 1), \quad (\alpha_2', \beta_2') = (\alpha_2, \beta_2 + 1).$$

Then the optimal posterior utility for the original instance equals

$$-c_{\text{sel}} + \gamma_2 f(1) + (1 - \gamma_2)(-c_{\text{sel}} + \gamma_1 f(1) + (1 - \gamma_1)X),$$

which can be rewritten as

$$-c_{\text{sel}}(2 - \gamma_2) + f(1)(\gamma_1 + \gamma_2 - \gamma_1\gamma_2) + (1 - \gamma_1)(1 - \gamma_2)X.$$

Consider the following alternative policy however. We first choose arm 1. If $r_1 = 1$ then we skip all remaining episodes. Otherwise, we choose arm 2. If $r_2 = 1$ we again skip all remaining episodes. If $r_1 = 2$ however, we play the optimal policy for the length $m - 2$ episode $(g', (\alpha'_i, \beta'_i)_{i=1}^2)$. The alternative policy has posterior utility equal to

$$-c_{\text{sel}}(2 - \gamma_1) + f(1)(\gamma_1 + \gamma_2 - \gamma_1\gamma_2) + (1 - \gamma_1)(1 - \gamma_2)X.$$

which exceeds the posterior utility of the posterior optimal policy by $c_{\text{sel}}(\gamma_1 - \gamma_2) > 0$, contradicting the initial optimality assumption. $\qquad\square$

Recall that $N_{\mathcal{P}} = \left\lfloor (m - 1)\frac{\beta_{2,0}}{\alpha_{2,0}} \right\rfloor + 1$. Define the arm1-stability event $\text{Ev}_1$ as in Equation (5) and $\text{Ev}_2$ as in Equation (10).

**Lemma E.2.** *Assume* $\text{Ev}_1$ *and* $\text{Ev}_2$ *hold. Assume further that either* $f(1) = f(m)$ *or* $f(0) = f(m - 1)$. *Then arm 2 is pulled at most* $N_{\mathcal{P}}$ *times.*

*Proof.* Suppose, for contradiction, that arm 2 is pulled $N_{\mathcal{P}} + 1$ times. Let $t$ be the round of the $(N_{\mathcal{P}} + 1)$-st pull. Since $\text{Ev}_1$ holds, Lemma C.2 implies

$$\gamma_{1,t-1} \geq \gamma_{1,0} - \Delta/4.$$

Invoking Claim D.6 we obtain

$$\frac{\alpha_{2,t-1} + m - 1}{\alpha_{2,t-1} + \beta_{2,t-1} + m - 1} \leq \gamma_{2,0}$$

Since $\gamma_{1,0} - \gamma_{2,0} = \Delta > 0$, it follows that

$$\frac{\alpha_{2,t-1} + m - 1}{\alpha_{2,t-1} + \beta_{2,t-1} + m - 1} \leq \gamma_{2,0} < \gamma_{1,0} - \Delta/4 \leq \gamma_{1,t-1}.$$

By Lemma E.1, arm 2 cannot be played in this round, contradicting the assumption that it is played more than $N_{\mathcal{P}}$ times. $\qquad\square$

We can now prove Theorem 4.6(b).

*Proof of Theorem 4.1(b).* The proof is similar to Theorem 4.6. As before, we have $\Pr[\text{Ev}_1] \geq \Delta/4$ by Lemma C.3. As before, define $\tau$ such that $\Pr[\mu_1 \notin (\tau, 1 - \tau)] = \Delta/16$ and note that $\tau > 0$ since $\alpha_{1,0}, \beta_{1,0} > 0$. Taking union bound, we conclude that

$$\Pr[\text{Ev}_1 \text{ and } \mu_1 \in (\tau, 1 - \tau)] \geq \Delta/8.$$

Define $\text{Ev}_3$ as $\mu_2 \in (1 - \kappa, 1 - \kappa/2)$ where $\kappa = \frac{\tau}{2}$. The event holds with probability $\Theta_{\mathcal{P}}(1) > 0$ as before since the density function for arm 2 is strictly positive. Conditioned on $\text{Ev}_3$, we have $\text{Ev}_2$ (see Equation (10)) with probability at least $(\kappa/2)^{N_{\mathcal{P}}}$ which is $(\Theta_{\mathcal{P}}(1))^m$. If all of these hold, then $\mu_2 \geq \mu_1 + c_{\mathcal{P}}$ and $\texttt{FAIL}$ happens. Concretely, we have $\mu_2 \geq 1 - \kappa = 1 - \tau + \kappa \geq \mu_1 + \kappa$. Setting $c_{\mathcal{P}} = \kappa/2$, we have $\mu_2 \geq \mu_1 + c_{\mathcal{P}}$. We also have $\mu_1 \in (c_{\mathcal{P}}, 1 - c_{\mathcal{P}})$ because $\mu_1 \in (\tau, 1 - \tau)$ and $\mu_2 \in (c_{\mathcal{P}}, 1 - c_{\mathcal{P}})$ because $\mu_2 \in (1 - \kappa, 1 - \kappa/2)$. Finally, since $\text{Ev}_2$ and $\text{Ev}_1$ hold, by Lemma E.2 we conclude that arm 2 is not played more than $N_{\mathcal{P}}$ times. Since arms 1 and 2 have independent priors and reward tapes, the probability of all of the mentioned events holding is at least $\Theta_{\mathcal{P}}(1)^m$ as required and the proof is complete. $\qquad\square$

# F. Proof of Theorem 4.3: $m = 2$, Arbitrary Aggregation Function

The proof follows the same overall structure as Theorem 4.6 but requires important changes that we discuss here. Firstly, in order to obtain a condition that ensures arm 2 is not pulled, we cannot rely on Appendix C.1 as it only holds for symmetric functions. Indeed, somewhat surprisingly, without some assumption on the cost function, the lemma is actually false (see Appendix G.2). To overcome this, we use a modified result as formalized by Lemma F.1. Importantly, the result also requires that the $\frac{\alpha_{1,t-1}}{\alpha_{1,t-1}+\beta_{1,t-1}+1} \neq \gamma_{2,t-1}$. As such, subsequent steps in the proof, appearing in Lemma C.4 in the previous proof, also become more involved to ensure the condition is met (see Lemma F.2).

**Lemma F.1.** *Assume that* $\frac{\alpha_{2,t-1}+1}{\alpha_{2,t-1}+\beta_{2,t-1}+1} < \gamma_{1,t-1}$ *and* $\frac{\alpha_{1,t-1}}{\alpha_{1,t-1}+\beta_{1,t-1}+1} \neq \gamma_{2,t-1}$. *Assume further that either* $c_{\text{sel}} < \gamma_{1,t-1} \cdot (f(1,1) - f(1,0))$ *or* $c_{\text{sel}} > f(0,1) - f(1,0)$. *Arm 2 will not be selected in round* $t$.

*Proof.* As in Lemma C.1, we assume that a deterministic posterior optimal policy plays arm 2 and use this to obtain a contradiction. Claim C.6 still holds in this setting as well since it does not use the symmetry of $f$. As such, we assume as before that $t$ is the first round of the episode. We begin by defining some notation which will be used throughout the rest of the proof. We define $\gamma_1 = \gamma_{1,t-1}$ and $\gamma_2 = \gamma_{2,t-1}$ for conveneince. Define $\gamma_1^+ = \frac{\alpha_{1,t-1}+1}{\alpha_{1,t-1}+\beta_{1,t-1}+1}$. To interpret this quantity, suppose we pull arm 1 once and observe the reward 1. The posterior distribution of arm 1's parameter then becomes $\text{Beta}(\alpha_{1,t-1} + 1, \beta_{1,t-1})$. The quantity $\gamma_1^+$ corresponds to precisely the expectation of this posterior distribution.

We similarly define $\gamma_1^- = \frac{\alpha_{1,t-1}}{\alpha_{1,t-1}+\beta_{1,t-1}+1}$ to be the expectation of the posterior of arm 1's parameter if we were to draw it once and observe the reward 0. The quantities $\gamma_2^+$ and $\gamma_2^-$ are defined similarly.

As before, Claim C.6 implies that arm 2 will not be played in the second round of the episode because, regardless of the value of $r_1$, we have $\gamma_{2,t} < \gamma_{1,t}$. This means there are 4 cases we need to consider for what happens in the second round based on the value of $r_1$.

- Case I: always skip.

- Case II: always play arm 1.

- Case III: if $r_1 = 0$, play arm 1; if $r_1 = 1$ skip.

- Case IV: if $r_1 = 1$, play arm 1; if $r_1 = 0$ skip.

We analyze each of these 4 cases separately. In each case, we obtain a contradiction by considering an alternative policy to the posterior optimal policy and show that its posterior utility exceeds that of the posterior optimal policy. Throughout, we use $\text{PU}_{\text{opt}}$ and $\text{PU}_{\text{alt}}$ to denote the posterior utility of the posterior optimal and alternative policies respectively,

**Case I.** In this case, consider the alternative policy of playing arm 1 and then skipping. Since both strategies always have $r_2 = 0$, we obtain

$$\text{PU}_{\text{opt}} = \Pr[r_1 = 1]f(1,0) + \Pr[r_1 = 0]f(0,0) = \gamma_1 f(1,0),$$

where we have used the fact that arm 1 is played to plug in the value of $\Pr[r_1 = 1]$ and the fact that $f(0,0) = 0$ to remove the second term. Similarly,

$$\text{PU}_{\text{opt}} = \Pr[r_1 = 1]f(1,0) + \Pr[r_1 = 0]f(0,0) = \gamma_2 f(1,0).$$

It follows that

$$\text{PU}_{\text{alt}} - \text{PU}_{\text{opt}} = (\gamma_1 - \gamma_2)(f(1,0) - f(0,0)),$$

which is always non-negative. If equality holds, we must have $f(1,0) = f(0,0)$. In this case however, the policy of always skipping both rounds would strictly improve as it does not pay the cost $c_{\text{sel}}$ in the first round.

**Case II.** We first establish the relationship between $\gamma_1^+, \gamma_1, \gamma_1^-$ given that both $\gamma_1^+$ and $\gamma_1^-$ effectively correspond to the conditional expectation of $\gamma_1$ under some event. Formally, assume we first sample some value $r$ from the distribution of arm 1 and then look at the expectation of arm 1's parameter conditioned on $r$. If $r = 1$, then the expectation equals $\gamma_1^+$ while if $r = 0$, the expectation equals $\gamma_1^-$. Moreover, the probability of observing $r = 1$ and $r = 0$ are $\gamma_1$ and $(1 - \gamma_1)$ respectively. If we were to not condition on the value of $r$ however, the expected value of arm 1's parameter is $\gamma_1$. By total expectation, it follows that

$$\begin{aligned}
\gamma_1 &= \mathbb{E}\left[\mu_1\right] \\
&= \Pr\left[r = 1\right]\mathbb{E}\left[\mu_1 \mid r = 1\right] + \Pr\left[r = 0\right]\mathbb{E}\left[\mu_1 \mid r = 0\right] \\
&= \gamma_1\gamma_1^+ + (1 - \gamma_1)\gamma_1^-
\end{aligned} \tag{12}$$

We divide the proof into two further sub-cases based on the relationship between $\gamma_1^-$ and $\gamma_2$. Specifically, recall that we have $\gamma_2 < \gamma_2^+ < \gamma_1$ by assumption; i.e., even if we see reward 1 from arm 2, we would not prefer it to arm 1. However, if we were to see the reward 0 from arm 1, then its expected parameter would drop from $\gamma_1$ to $\gamma_1^-$ which may potentially be lower than $\gamma_2$. We separately hande the cases $\gamma_1^- < \gamma_2$ and $\gamma_1^- > \gamma_2$. Note that $\gamma_1^- \neq \gamma_2$ by assumption of the lemma.

We start with the case $\gamma_1^- > \gamma_2$. In this case, consider the alternative policy that plays arm 1 in both rounds. We compare the posterior utility of the posterior optimal policy with this new alternative policy. We start with the posterior optimal policy. The value of $r_1$ is sampled from the Bernoulli distribution with parameter $\gamma_2$ and $r_2$ is sampled from the Bernoulli distribution with parameter $\gamma_1$. Since the two arms are independent, the values are sampled independently which means $\text{PU}_{\text{opt}}$ equals

$$\begin{aligned}
&- 2c_{\text{sel}} + \sum_{(a,b)\in\{0,1\}^2} f(a,b)\Pr\left[(r_1, r_2) = (a,b)\right] \\
&= -2c_{\text{sel}} + \sum_{(a,b)\in\{0,1\}^2} f(a,b)\Pr\left[r_1 = a\right]\Pr\left[r_2 = b\right] \\
&= -2c_{\text{sel}} + \gamma_2\gamma_1 f(1,1) + \gamma_2(1 - \gamma_1)f(1,0) + (1 - \gamma_2)\gamma_1 f(0,1) + (1 - \gamma_2)(1 - \gamma_1)f(0,0).
\end{aligned}$$

where for the last equality we have plugged in the values of $f(a,b)$ (note that $f(0,0) = 0$ by assumption).

For the alternative policy, both draws are from arm 1. While the two draws are independent if we condition on the value of arm 1's parameter, this value is not known; as such, the value we observe for the first draw would cause an update in the posterior distribution, affecting the second draw. More specifically, $r_1$ is sampled from the Bernoulli distribution with parameter $\gamma_1$ and, depending on whether $r_1 = 1$ or $r_1 = 0$, the value of $r_2$ is sampled from a Bernoulli with parameter equal to either $\gamma_1^+$ or $\gamma_1^-$. As such, the posterior utility of the policy equals

$$-2c_{\text{sel}} + \gamma_1\gamma_1^+ f(1,1) + \gamma_1(1 - \gamma_1^+)f(1,0) + (1 - \gamma_1)\gamma_1^- f(0,1) + (1 - \gamma_1)(1 - \gamma_1^-)f(0,0).$$

In order to better compare with the posterior utility of the posterior optimal policy however, we consider a "backwards" view to this process. We first sample the value of $r_2$ and then sample the value of $r_1$ based on the updated posterior. This simplifies our analysis because both strategies sample arm 1 in the second round; by realizing the value of $r_2$ first, we can more easily compare the strategies as $r_2$ would now be sampled from the same distribution. Formally, since $r_2$ is sampled from a Bernoulli with parameter $\gamma_1$ and, depending on the value of arm $r_2$, the value of $r_1$ is sampled from Bernoulli with parameter either $\gamma_1^+$ or $\gamma_1^-$, the posterior utility of the alternative policy $\text{PU}_{\text{alt}}$ can be written as [11]

$$-2c_{\text{sel}} + \gamma_1\gamma_1^+ f(1,1) + \gamma_1(1 - \gamma_1^+)f(0,1) + (1 - \gamma_1)\gamma_1^- f(1,0) + (1 - \gamma_1)(1 - \gamma_1^-)f(0,0).$$

It follows that

$$\text{PU}_{\text{alt}} - \text{PU}_{\text{opt}} = \gamma_1(\gamma_1^+ - \gamma_2)(f(1,1) - f(0,1)) + (1 - \gamma_1)(\gamma_1^- - \gamma_2)(f(1,0) - f(0,0)).$$

Since $f(.,.)$ is never decreasing in its coordinates and $\gamma_1^+, \gamma_1^- > \gamma_2$, we have $\text{PU}_{\text{alt}} \geq \text{PU}_{\text{opt}}$. Moreover, if $\text{PU}_{\text{opt}} = \text{PU}_{\text{alt}}$, then we must have $f(1,1) = f(0,1)$ and $f(1,0) = f(0,0)$; in other words, the value of $r_1$ does not matter. In this case, the

---

[11] One can alternatively obtain this alternative characterization using the initial characterization and applying Equation (12).

utility of both strategies can be simplified as

$$-2c_{\text{sel}} + \gamma_1 f(1,1) + (1-\gamma_1)f(0,0).$$

Note however that if we were to skip the first round and play arm 1 in the secound round, then the posterior utility would equal

$$-c_{\text{sel}} + \gamma_1 f(1,1) + (1-\gamma_1)f(0,0),$$

which is strictly higher since $c_{\text{sel}} > 0$. Therefore we have a contradiction as the posterior optimal policy is not maximizing posterior utility, finishing the proof in this case.

We next handle the case that $\gamma_1^- < \gamma_2$. In this case, we consider a different alternative policy: we choose arm 1 in the first round and choose arm 1 again in the second round if $r_1 = 1$, choosing arm 2 instead if $r_1 = 0$. The posterior utility of this policy equals

$$-2c_{\text{sel}} + \gamma_1(\gamma_1^+ f(1,1) + (1-\gamma_1^+)f(1,0)) + (1-\gamma_1)(\gamma_2 f(0,1) + (1-\gamma_2)f(0,0)).$$

Recall however that $f(0,0) = 0$ by assumption. Taking the difference with $\text{PU}_{\text{opt}}$,

$$
\begin{aligned}
\text{PU}_{\text{alt}} - \text{PU}_{\text{opt}} &= f(0,1)((1-\gamma_1)\gamma_2 - (1-\gamma_2)\gamma_1) + f(1,1)\gamma_1(\gamma_1^+ - \gamma_2) + f(1,0)(\gamma_1(1-\gamma_1^+) - \gamma_2(1-\gamma_1)) \\
&= -f(0,1)(\gamma_1 - \gamma_2) + f(1,1)\gamma_1(\gamma_1^+ - \gamma_2) + f(1,0)(\gamma_1(1-\gamma_1^+) - \gamma_2(1-\gamma_1)) \\
&= -f(0,1)(\gamma_1 - \gamma_2) + f(1,1)\gamma_1(\gamma_1^+ - \gamma_2) + f(1,0)(\gamma_1 - \gamma_2) - f(1,0)\gamma_1(\gamma_1^+ - \gamma_2) \\
&= (\gamma_1 - \gamma_2)(f(1,0) - f(0,1)) + \gamma_1(\gamma_1^+ - \gamma_2)(f(1,1) - f(1,0))
\end{aligned}
$$

Recall however that $\gamma_2 > \gamma_1^-$ by assumption. Plugging in Equation (12), this implies

$$\gamma_1\gamma_1^+ + (1-\gamma_1)\gamma_2 > \gamma_1,$$

which can be rewritten as

$$\gamma_1(\gamma_1^+ - \gamma_2) > \gamma_1 - \gamma_2.$$

Therefore, since $f(1,1) \geq f(1,0)$,

$$
\begin{aligned}
\text{PU}_{\text{alt}} - \text{PU}_{\text{opt}} &\geq (\gamma_1 - \gamma_2)(f(1,0) - f(0,1)) + (\gamma_1 - \gamma_2)(f(1,1) - f(1,0)) \\
&= (\gamma_1 - \gamma_2)(f(1,0) - f(0,1) + f(1,1) - f(1,0)) \\
&= (\gamma_1 - \gamma_2)(f(1,1) - f(0,1))
\end{aligned}
$$

which is always non-negative. Furthermore, in order to have equality, we must have $f(1,1) = f(0,1)$ and $f(1,1) = f(1,0)$; the latter inequality needs to hold since we lower bounded $\gamma_1(\gamma_1^+ - \gamma_2)$ with $\gamma_1 - \gamma_2$. If $f(1,1) = f(1,0)$ however, consider the slight alteration of the alternative policy we considered: if $r_1 = 1$ then skip the second round. With probability $\gamma_1$, this would save a cost of $c_{\text{sel}}$ for exploring the second round. Therefore, the posterior optimal policy is not maximizing posterior utility, giving us the contradiction we wanted.

**Case III.** By assumption of the lemma, we have either $c_{\text{sel}} < \gamma_1(f(1,1) - f(1,0))$ or $c_{\text{sel}} > f(0,1) - f(1,0)$.

In the first case, observe that if $r_1 = 1$, then it is better to play arm 1 than to skip. Since $\gamma_2 > 0$, this means the policy that plays arm 1 instead of skipping when $r_1 = 1$ would improve on the posterior optimal policy.

In the second case, observe that by definition, we have

$$\text{PU}_{\text{opt}} = -c_{\text{sel}} + \gamma_2 f(1,0) + (1-\gamma_2)(-c_{\text{sel}} + \gamma_1 f(0,1)).$$

Consider the alternative policy of picking arm 1 first and picking arm 2 if $r_1 = 0$ an skipping in the second round otherwise. We have

$$\text{PU}_{\text{alt}} = -c_{\text{sel}} + \gamma_1 f(1,0) + (1-\gamma_1)(-c_{\text{sel}} + \gamma_2 f(0,1)).$$

It follows that

$$\text{PU}_{\text{alt}} - \text{PU}_{\text{opt}} = (\gamma_1 - \gamma_2)(f(1,0) + c_{\text{sel}} - f(0,1)).$$

Since $c_{\text{sel}} \geq f(0,1) - f(1,0)$ and $\gamma_1 > \gamma_2$, we have $\text{PU}_{\text{alt}} > \text{PU}_{\text{opt}}$, finishing the proof.

**Case IV.** In this case, consider the alternative policy that plays arm 1 in the first round and then skips if $r_1 = 0$ and plays arm 1 again otherwise. The policy has posterior utility

$$-c_{\text{sel}} + \gamma_1(-c_{\text{sel}} + \gamma_1^+ f(1,1) + (1 - \gamma_1^+)f(1,0)) + (1 - \gamma_1)f(0,0),$$

while the posterior optimal policy has posterior utility

$$-c_{\text{sel}} + \gamma_2(-c_{\text{sel}} + \gamma_1 f(1,1) + (1 - \gamma_1)f(1,0)) + (1 - \gamma_2)f(0,0).$$

We first claim that

$$-c_{\text{sel}} + \gamma_1 f(1,1) + (1 - \gamma_1)f(1,0) > 0.$$

If this is not the case, then the posterior utility of the posterior posterior optimal policy would be at most $-c_{\text{sel}}$; this is not possible however since skipping both rounds would give the utility $0$ which is strictly better. Since $\gamma_1 > \gamma_2$, this means that

$$\text{PU}_{\text{opt}} < -c_{\text{sel}} + \gamma_1(-c_{\text{sel}} + \gamma_1 f(1,1) + (1 - \gamma_1)f(1,0)).$$

Subtracting from $\text{PU}_{\text{alt}}$ we obtain

$$\text{PU}_{\text{alt}} - \text{PU}_{\text{opt}} > \gamma_1(\gamma_1^+ - \gamma_1)(f(1,1) - f(1,0)) \geq 0,$$

finishing the proof. $\qquad\square$

Now, define the arm1-stability event $\text{Ev}_1$ as in Equation (5). Set $N_0 = \left\lfloor \frac{\beta_{2,0}}{\alpha_{2,0}} \right\rfloor + 1$ and $N_1 = \left\lfloor \frac{\gamma_{2,0}}{3\Delta/4} \right\rfloor + 1$. Define

$$\text{Ev}_2 = \left\{ \texttt{tape}_{2,\ell} = 0 \text{ for all } \ell \in [N_0 + N_1] \right\}.$$

**Lemma F.2.** *Assume that $\text{Ev}_1$ and $\text{Ev}_2$ hold. Assume further that either $c_{\text{sel}} < (\gamma_{1,0} - \Delta/4) \cdot (f(1,1) - f(1,0))$ or $c_{\text{sel}} > f(0,1) - f(1,0)$. Arm 2 is not selected more than $N_0 + N_1$ times.*

*Proof.* Define $\gamma_1^{-,n}$ as $\frac{\alpha_{1,0} + \sum_{\ell \leq n} \texttt{tape}_{1,\ell}}{\alpha_{1,0} + \beta_{1,0} + n + 1}$. Define the set $\texttt{badset}$ as

$$\texttt{badset} = \left\{ \gamma_1^{-,n} : \gamma_1^{-,n} \leq \gamma_{2,0} \right\}$$

Recall that

$$\gamma_2^{(n)} = \frac{\alpha_{2,0} + \sum_{\ell \leq n} \texttt{tape}_{2,\ell}}{\alpha_{2,0} + \beta_{2,0} + n}.$$

By definition of $\text{Ev}_2$, this implies that

$$\gamma_2^{(n)} = \frac{\alpha_{2,0}}{\alpha_{2,0} + \beta_{2,0} + n} \text{ for all } n \leq N_0 + N_1.$$

We first claim that there exists some $n \in [N_0, N_0 + N_1]$ such that $\gamma_2^{(n)} \notin \texttt{badset}$. Assume that this is not the case; i.e., $\gamma_2^{(n)} \in \texttt{badset}$ for all $n \in [N_0, N_0 + N_1]$. Since $\gamma_2^{(n)}$ are distinct for $n$, it follows that

$$N_1 + 1 = \left| \left\{ \gamma_2^{(n)} : n \in [N_0, N_0 + N_1], \right\} \right| \leq |\texttt{badset}|.$$

Take some $n \geq N_1$ such that $\gamma_1^{-,n} \in \texttt{badset}$. Such an $n$ must exists because otherwise we would have $|\texttt{badset}| \leq N_1$ (note that we also consider $n = 0$ when forming $\texttt{badset}$). By definition of $\texttt{badset}$, we have $\gamma_1^{-,n} \leq \gamma_{2,0}$. This means $\frac{\alpha_{1,0} + \sum_{\ell \leq n} \texttt{tape}_{1,\ell}}{\alpha_{1,0} + \beta_{1,0} + n + 1} \leq \gamma_{2,0}$. Note however that $\frac{\alpha_{1,0} + \sum_{\ell \leq n} \texttt{tape}_{1,\ell}}{\alpha_{1,0} + \beta_{1,0} + n} \geq \gamma_{1,0} - \Delta/4$ by $\text{Ev}_1$. Therefore,

$$\frac{\alpha_{1,0} + \beta_{1,0} + n + 1}{\alpha_{1,0} + \beta_{1,0} + n} \geq \frac{\gamma_{1,0} - \Delta/4}{\gamma_{2,0}}.$$

It follows that

$$\frac{1}{\alpha_{1,0} + \beta_{1,0} + n} \geq \frac{\gamma_{1,0} - \Delta/4 - \gamma_{2,0}}{\gamma_{2,0}} = \frac{3\Delta/4}{\gamma_{2,0}}.$$

Therefore,

$$n \leq \frac{\gamma_{2,0}}{3\Delta/4}.$$

Since $n \geq N_1$, it follows that $N_1 \leq \frac{\gamma_{2,0}}{3\Delta/4}$. This contradicts the definition of $N_1$. Therefore, the initial assumption was incorrect and $\gamma_2^{(n)} \notin \texttt{badset}$ for some $n \in [N_0, N_0 + N_1]$.

We next claim that if $n \in [N_0, N_0 + N_1]$ and $\gamma_2^{(n)} \notin \texttt{badset}$, then arm 2 will not be played more than $n$ times. Assume for the sake of contradiction that it is and let $t$ be the round at which arm 2 is played for the $(n + 1)$-th time. We claim that the preconditions Lemma F.1 holds.

- We need to show that $\frac{\alpha_{2,t-1}+1}{\alpha_{2,t-1}+\beta_{2,t-1}+1} < \gamma_{1,t-1}$. By Claim C.5, since $n \geq N_0 > \frac{\beta_{2,0}}{\alpha_{2,0}}$, we have $\frac{\alpha_{2,t-1}+1}{\alpha_{2,t-1}+\beta_{2,t-1}+1} \leq \gamma_{2,0}$. By $\mathrm{Ev}_1$ and Lemma C.2 we have $\gamma_{1,t-1} \geq \gamma_{1,0} - \Delta/4 > \gamma_{2,0}$ and we are done.

- Letting $\gamma_{1,t-1}^-$ denote $\frac{\alpha_{1,t-1}}{\alpha_{1,t-1}+\beta_{1,t-1}+1}$, we need to show that $\gamma_{1,t-1}^- \neq \gamma_{2,t-1}$. Assume for contradiction that $\gamma_{1,t-1}^- = \gamma_{2,t-1}$. Since arm 2 has been pulled at most $n \leq N_0 + N_1$ times and $\texttt{tape}_{2,\ell} = 0$ for $\ell \leq N_0 + N_1$ by $\mathrm{Ev}_2$, we have $\gamma_{2,t-1} \leq \gamma_{2,0}$. This means that $\gamma_{1,t-1}^- \leq \gamma_{2,0}$ which in turn means that $\gamma_{1,t-1}^- \in \texttt{badset}$. Since arm 2 has been pulled exactly $n$ times before, it follows that $\gamma_2^{(n)} = \gamma_{2,t-1} = \gamma_{1,t-1}^- \in \texttt{badset}$ which contradicts the choice of $n$.

- We need to show that either $c_{\mathrm{sel}} < \gamma_{1,t-1}(f(1,1) - f(1,0))$ or $c_{\mathrm{sel}} > f(0,1) - f(1,0)$. By the lemma statement, we have either $c_{\mathrm{sel}} < (\gamma_{1,0} - \Delta/4) \cdot (f(1,1) - f(1,0))$ or $c_{\mathrm{sel}} > f(0,1) - f(1,0)$. Since $\gamma_{1,t-1} \geq \gamma_{1,0} - \Delta/4$ because of $\mathrm{Ev}_1$ and Lemma C.2, the claim follows.

By Lemma F.1, arm 2 cannot be pulled at round $t$ and the proof is complete. $\qquad\square$

This leads us to the following theorem.

*Proof of Theorem 4.3.* The proof is similar to that of Theorem 4.6. As before, we have $\Pr[\mathrm{Ev}_1] \geq \Delta/4$. Setting an appropriate $\tau > 0$ we have $\Pr[\mu_1 \notin (\tau, 1 - \tau)] \leq \Delta/16$. It follows that with probability at least $\Delta/8$ we have $\mathrm{Ev}_1$ and $\mu_1 \in (\tau, 1 - \tau)$.

Set $\kappa = \tau/2$ as before. As before we define $\mathrm{Ev}_3$ to be the event that $\mu_2 \in (1 - \kappa, 1 - \kappa/2)$. The probability of this is strictly positive (since the density function is strictly positive) and therefore it is at least $\Theta_P(1)$. Conditioned on $\mathrm{Ev}_3$ holding, $\mathrm{Ev}_2$ holds with probability at least $\kappa^{N_0 + N_1}$. Since $\kappa$ is decided based on the prior and so are $N_0, N_1$, this probability is $\Theta_P(1)$. So with probability $\Theta_{\mathcal{P}}(1)$ we have both $\mathrm{Ev}_3$ and $\mathrm{Ev}_2$ as well.

Now, since the priors were independent, we have $\mathrm{Ev}_1, \mathrm{Ev}_2, \mathrm{Ev}_3$ and $\mu_1 \in (\tau, 1 - \tau)$ with probability $\Theta_{\mathcal{P}}(1)$. Setting $c_{\mathcal{P}} = \kappa/2$ (note that $\kappa$ depends only on the prior) we have $\mu_2 \geq \mu_1 + c_{\mathcal{P}}$ and $\mathrm{Ev}_1, \mathrm{Ev}_2$ hold. We also have $\mu_1 \in (c_{\mathcal{P}}, 1 - c_{\mathcal{P}})$ because $\mu_1 \in (\tau, 1 - \tau)$ and $\mu_2 \in (c_{\mathcal{P}}, 1 - c_{\mathcal{P}})$ because $\mu_2 \in (1 - \kappa, 1 - \kappa/2)$.

Finally, the precondition of Lemma F.2 on $c_{\mathrm{sel}}$ also holds by the assumption in the theorem. Specifically, either $f(1,1) > f(1,0)$, in which case $c_{\mathrm{sel}} < c_{\max} = (\gamma_{1,0} - \Delta/4)$, or $f(1,1) = f(1,0)$, in which case $c_{\mathrm{sel}} > 0 = f(1,1) - f(1,0) \geq f(0,1) - f(1,0)$. It follows that arm 2 is not pulled more than $N_0 + N_1$ times, finishing the proof. $\qquad\square$

# G. Examples

### G.1. Comparison with single round episodes

In this section, we demonstrate via a simple example why, unlike the single round episodes, the agents do not always pick the arm that has the highest current posterior mean.

Consider the following instance.

$$(\alpha_1, \beta_1) = (2, 9), \qquad (\alpha_2, \beta_2) = (1, 5).$$

We set $f(r_1, r_2) = \min(r_1, r_2)$ and $c_{\text{sel}} = 10^{-3}$.

It is clear that $\mathbb{E}[\mu_1] = \frac{2}{11} > \frac{1}{6} = \mathbb{E}[\gamma_2]$. However, a myopic agent will play arm 2 on the first round and play arm 2 in the second instance if the initial reward is 0 and play skip otherwise.

To see why, observe that if we were to play arm 2 once and observe the reward 1, then its posterior value would be update to $\frac{\alpha_2 + 1}{\alpha_2 + \beta_2 + 1} = \frac{2}{7}$. As such, the probability of observing two draws of 1 by pulling arm 2 is $\frac{1}{6} \cdot \frac{2}{7} = \frac{1}{22}$. In contrast, if we were to pull arm 1 once and see a reward of 1, then its posterior mean would equal $\frac{3}{12} = \frac{1}{4}$, which means the probability of both rewards being 1 is $\frac{2}{11} \cdot \frac{1}{4} = \frac{1}{22}$. It follows that choosing arm 2 in the first round and then re-choosing it if we see a reward of 1 (skipping otherwise) is strictly preferable to following a similar strategy with arm 1. Note that, despite the positive cost, skipping from the beginning is not desirable given the low cost.

### G.2. Hard instance for non-symmetric functions

In this section we show that without the assumption on cost, Lemma F.1 does not hold in general. Note that this is in contrast to Lemma C.1 for symmetric functions which does not require an assumption on cost.

Consider the following instance.

$$(\alpha_1, \beta_1) = (2.54375, \ 2.08125) \quad \text{and} \quad (\alpha_2, \beta_2) = (450, \ 550).$$

This implies that

$$\gamma_1 = \frac{\alpha_1}{\alpha_1 + \beta_1} = 0.55 \quad \text{and} \quad \gamma_2 = \frac{\alpha_2}{\alpha_2 + \beta_2} = 0.45.$$

Moreover, letting

$$\gamma_i^+ := \frac{\alpha_i + 1}{\alpha_i + \beta_i + 1} \quad \text{and} \quad \gamma_i^- := \frac{\alpha_i}{\alpha_i + \beta_i + 1},$$

we have

$$\gamma_1^+ = \frac{2.54375 + 1}{2.54375 + 2.08125 + 1} = 0.63, \quad \gamma_1^- = \frac{2.54375}{2.54375 + 2.08125 + 1} \approx 0.45, \quad \text{and} \quad \gamma_2^+ = \frac{450 + 1}{1000 + 1} \approx 0.45055.$$

Set $c_{\text{sel}} = 0.13$. Define the function $f$ as

$$f(0, 0) = 0, \quad f(1, 0) = 1, \quad f(0, 1) = 1.2, \quad f(1, 1) = 1.2$$

We claim that the posterior optimal policy is as follows. In the first round choose arm 2 and observe reward $r_1$. If $r_1 = 0$, then choose arm 1 in the second round. Otherwise, skip the second round.

To see why, note the that the posterior utility of this policy equals

$$-0.13 + 0.45 \times 1 + 0.55 \times (-0.13 + 0.55 \times 1.2) = 0.6115$$

If we were to skip in the first round, then either we skip in the second round as well, obtaining a posterior utility of 0, or we play arm 1 or arm 2. Since $\gamma_1 \geq \gamma_2$, playing arm 1 would be better and the posterior utility in this case equals

$$-0.13 + 0.55 \times 1.2 = 0.53 < 0.6115.$$

As such, a posterior optimal policy cannot skip in the first round.

Next consider policies playing arm 1 in the first round. If $r_1 = 1$, then the policy should skip because the expected gain from playing arm 1 is at most $0.63 \times (1.2 - 1.0) = 0.126$ which is less than the cost $0.13$. If $r_1 = 0$ however, the policy should play arm 2 because the expected gain equals $0.45 \times 1.2 = 0.54$ exceeding the cost. Note that playing arm 1 would also lead to the same utility: since $r_1 = 0$, the probability of $r_1 = 1$ is $\gamma_1^- = 0.45$. Therefore, if a policy chooses arm 1 in the first round, then its posterior utility is maximized by choosing skip for $r_1 = 1$ and either arm 1 or arm 2 for $r_1 = 0$. The posterior utility of this policy equals

$$-0.13 + 0.55 \times 1 + 0.45 \times (-0.13 + 0.45 \times 1.2) = 0.6045 < 0.6115.$$

As such, the posterior optimal policy cannot play arm 1 in the first round either.

Finally, we consider all policies choosing arm 2 in the first round. If $r_1 = 1$, then the expected gain from playing arm 1 is at most $0.55 \times (1.2 - 1) = 0.11$ which is less than the cost $0.13$. The expected gain from playing arm 2 is even lower since $\gamma_2^+ < \gamma_1$. As such, after observing $r_1 = 1$ the policy should skip. As for $r_1 = 0$, the expected gain from playing arm 1 is $0.55 \times 1.2 = 0.66$ which far exceeds the cost of $0.13$. Therefore, the optimal policy is as described.

# H. Extension to $K > 2$ Arms

We extend Theorem 4.1 to instances with $K > 2$ arms. The key observation is that if every arm beyond the first two looks sufficiently worse than arm 2 according to the prior, the extra arms are never selected under the same events used in the two-arm analysis, so the proof carries through essentially unchanged.

**Setup.** Fix an EpiBSL instance with $K \geq 2$ arms, episode length $m \geq 2$, and symmetric aggregation function $f$. We retain the notation of Appendix D: arms 1 and 2 are the *main* arms with prior means $\gamma_{1,0} > \gamma_{2,0}$ and gap $\Delta = \gamma_{1,0} - \gamma_{2,0} > 0$. Each additional arm $a \in \{3, \ldots, K\}$ has a $\mathrm{Beta}(\alpha_a, \beta_a)$ prior with prior mean $\gamma_{a,0} = \alpha_a/(\alpha_a + \beta_a)$. We write $\alpha_{a,t}$ and $\beta_{a,t}$ for the posterior parameters at the start of round $t$ (so $\alpha_{a,0} = \alpha_a$, $\beta_{a,0} = \beta_a$).

Because arms $3, \ldots, K$ may themselves be high-quality, bounding arm 2's pull count alone does not establish failure: the agent could be doing well by pulling one of those arms instead. We therefore use the following strengthened failure event.

**Definition H.1.** Fix $c > 0$, $N \in \mathbb{N}$, and $K \geq 3$. Event $\mathrm{FAIL}_{c,N}$ occurs if all of the following hold:

- *(Bounded mean rewards)* $\mu_a \in (c, 1 - c)$ for every $a \in [K]$.

- *(Arm 2 dominates)* $\mu_2 \geq \mu_a + c$ for every $a \neq 2$.

- *(Bounded pulls of arm 2)* Arm 2 is pulled at most $N$ times.

- *(Arms $3, \ldots, K$ never pulled)* Each arm $a \in \{3, \ldots, K\}$ is pulled zero times.

Together, these conditions unambiguously describe failure: the agent is stuck on arm 1 despite arm 2 being demonstrably better than every other arm, and it never explores arms $3, \ldots, K$ at all. For $K = 2$, condition (iv) is vacuous and conditions (i)–(iii) reduce to Definition 3.1, so Definition H.1 is a strict generalization of the two-arm failure event.

**Theorem H.2.** *Consider the setup above. Suppose further that every arm $a \in \{3, \ldots, K\}$ satisfies*

$$\frac{\alpha_a + m - 1}{\alpha_a + \beta_a + m - 1} < \gamma_{2,0}. \tag{13}$$

*Then Property (P1) holds, with $\mathrm{FAIL}_{c_{\mathcal{P}}, N_{\mathcal{P}}}$ referring to Definition H.1.*

The following corollary derives linear Bayesian regret via the same failure $\to$ strong failure $\to$ regret chain as in Section 3; the proof is in Appendix H.2.

**Corollary H.3.** *Under the conditions of Theorem H.2, there exist constants $c_0 > 0$ and $N_0, c_{\max} < \infty$ depending on $\mathcal{P}$, $f$, $m$, and $K$, such that small enough cost $c_{\mathrm{sel}} < c_{\max}$ implies*

$$\mathrm{BReg}(T) \geq c_0 \cdot (T - N_0) \qquad \text{for any episode } T.$$

*This is the same form as Corollary 4.5.*

## H.1. Proof of Theorem H.2

We prove Theorem H.2. The proof mirrors that of Theorem 4.1 step by step; we record the three supporting lemmas that generalize the corresponding lemmas from Appendix D and indicate the (minor) changes in each proof.

### Step 1: Extending Lemma D.2.

**Lemma H.4.** *Assume $f : \{0, 1\}^m \to \mathbb{R}$ is symmetric, coordinatewise increasing, and $f(m - 1) = f(0)$. Assume further that for every arm $a \in \{2, \ldots, K\}$,*

$$\frac{\alpha_{a,t-1} + m - 1}{\alpha_{a,t-1} + \beta_{a,t-1} + m - 1} < \gamma_{1,t-1}. \tag{14}$$

*Then any deterministic posterior optimal policy has one of the following two forms:*

- *Skip all rounds.*

- *Start by playing arm* 1 *and continue doing so while observed rewards equal* 1*; skip all remaining rounds upon observing a reward of* 0.

*Proof.* The proof is by induction on $m$, exactly as in Lemma D.2. The only difference is that the policy may start with any arm $a \in \{2, \ldots, K\}$, not just arm 2.

The induction structure is unchanged. Condition (14) for arm $a$ plays the same role as the corresponding condition on arm 2 in the original proof: it ensures that (i) when the induction hypothesis is applied to the sub-episode after a first pull, the precondition still holds (the same monotonicity argument: $\alpha_{a,t} \leq \alpha_{a,t-1} + 1$ and $\beta_{a,t} \geq \beta_{a,t-1}$), and (ii) the sequence of pulls must be $[a, 1, \ldots, 1]$ if the policy starts with arm $a$.

To rule out starting with arm $a \geq 2$, the same utility comparison applies: the comparison in Lemma D.2 uses only $\gamma_{1,t-1} > \gamma_{a,t-1}$. Since $\gamma_{a,t-1} = \frac{\alpha_{a,t-1}}{\alpha_{a,t-1}+\beta_{a,t-1}} \leq \frac{\alpha_{a,t-1}+m-1}{\alpha_{a,t-1}+\beta_{a,t-1}+m-1} < \gamma_{1,t-1}$ (by (14)), the comparison is arm-$a$-agnostic and carries through for all $a \geq 2$. $\qquad\square$

## Step 2: Extending Lemma D.3.

**Lemma H.5.** *Assume $f : \{0,1\}^m \to \mathbb{R}$ is symmetric and coordinatewise increasing. Assume Equation* (7) *holds. Suppose that for every arm $a \in \{2, \ldots, K\}$,*

$$\frac{\alpha_{a,t-1} + m - 1}{\alpha_{a,t-1} + \beta_{a,t-1} + m - 1} < \gamma_{1,t-1}. \tag{15}$$

*Then no arm from $\{2, \ldots, K\}$ is selected in round $t$.*

*Proof.* The proof is similar to that of Lemma D.3, with arm 2 replaced by an arbitrary arm $a \in \{2, \ldots, K\}$.

The four cases (I: always skip; II: always arm 1; III: arm 1 if $r_1 = 0$; IV: arm 1 if $r_1 = 1$) are resolved by the same arguments:

- Cases I, II, and III depend only on $\gamma_{1,t-1} > \gamma_{a,t-1}$ and on Equations (7) to (9); neither of these is arm-2-specific.

- Case IV invokes Lemma D.2 in the original proof; here it invokes Lemma H.4 instead.

Since none of the arguments require the non-arm-1 arm to be specifically arm 2, the conclusion holds for all $a \in \{2, \ldots, K\}$. $\qquad\square$

## Step 3: Extending Lemma D.5.

**Lemma H.6.** *Adopt the notation and assumptions of Lemma D.5 (events $\mathrm{Ev}_1$, $\mathrm{Ev}_2$, and Equation* (11)*). Assume additionally that condition (13) holds for every arm $a \in \{3, \ldots, K\}$. Then:*

(a) *Arm 2 is pulled at most $N_{\mathcal{P}}$ times (same bound as Lemma D.5).*

(b) *Arms $3, \ldots, K$ are never pulled.*

*Proof.* **Part (a).** Similar to Lemma D.5: after $N_{\mathcal{P}}$ all-zero pulls of arm 2, its optimistic posterior mean $\frac{\alpha_{2,t-1}+m-1}{\alpha_{2,t-1}+\beta_{2,t-1}+m-1}$ drops below $\gamma_{2,0}$, and event $\mathrm{Ev}_1$ gives $\gamma_{1,t-1} \geq \gamma_{1,0} - \Delta/4 > \gamma_{2,0}$, so Lemma H.5 prohibits a further pull of arm 2.

**Part (b).** Consider any round $t$. Since arm $a \in \{3, \ldots, K\}$ has never been pulled, $\alpha_{a,t-1} = \alpha_a$ and $\beta_{a,t-1} = \beta_a$. By condition (13) and event $\mathrm{Ev}_1$:

$$\frac{\alpha_a + m - 1}{\alpha_a + \beta_a + m - 1} < \gamma_{2,0} < \gamma_{1,0} - \frac{\Delta}{4} \leq \gamma_{1,t-1}.$$

Hence condition (15) holds for arm $a$, and Lemma H.5 prohibits selecting arm $a$ in round $t$. Since this holds for every $t$, arm $a$ is never pulled. $\qquad\square$

**Proof of Theorem H.2.**

*Proof.* We use the same $c_{\max}$, $N_{\mathcal{P}}$, $c_{\mathcal{P}}$, and events $\texttt{Ev}_1$, $\texttt{Ev}_2$ as in the proof of Theorem 4.1 (defined only for arms 1 and 2).

**Conditions (iii)–(iv) of** $\texttt{FAIL}$ (bounded pulls of arm 2; arms $3, \ldots, K$ never pulled) follow from Lemma H.6, exactly as in the original proof.

**Conditions (i)–(ii) for arms** 1 **and** 2 (bounded mean rewards; arm 2 dominates arm 1) follow by the same argument as in the proof of Theorem 4.1.

**Conditions (i)–(ii) for arms** $3, \ldots, K$**.** For each $a \in \{3, \ldots, K\}$, define the event $\texttt{Ev}_a^+ := \{\mu_a \in (\tau, 1 - \tau)\}$, where $\tau$ is the same prior-determined constant as in the proof of Theorem 4.1. Each $\texttt{Ev}_a^+$ satisfies $\Pr[\texttt{Ev}_a^+] = \Theta_{\mathcal{P}}(1) > 0$ and is independent of the events for arms 1 and 2. Under $\texttt{Ev}_3$ we have $\mu_2 \geq 1 - \kappa$, and under $\texttt{Ev}_a^+$ we have $\mu_a \leq 1 - \tau = 1 - 2\kappa$, so

$$\mu_2 - \mu_a \geq (1 - \kappa) - (1 - 2\kappa) = \kappa = 2c_{\mathcal{P}} > c_{\mathcal{P}},$$

giving condition (ii) for arm $a$. Condition (i) holds since $\mu_a \in (\tau, 1 - \tau) \subset (c_{\mathcal{P}}, 1 - c_{\mathcal{P}})$.

**Probability.** As in the proof of Theorem 4.1, four events control the arms 1 and 2 part of $\texttt{FAIL}$:

(i) $\texttt{Ev}_1$ (stability of arm 1): $\Pr[\texttt{Ev}_1] \geq \Delta/4$ by Lemma C.3.

(ii) $\{\mu_1 \in (\tau, 1 - \tau)\}$: combined with $\texttt{Ev}_1$ via union bound, $\Pr[\texttt{Ev}_1 \text{ and } \mu_1 \in (\tau, 1 - \tau)] \geq \Delta/8$.

(iii) $\texttt{Ev}_3 = \{\mu_2 \in (1 - \kappa, 1 - \kappa/2)\}$: probability $\Theta_{\mathcal{P}}(1) > 0$ since the prior density of arm 2 is strictly positive.

(iv) $\texttt{Ev}_2$ (arm 2's tape all zeros for the first $N_{\mathcal{P}}$ pulls): $\Pr[\texttt{Ev}_2 \mid \texttt{Ev}_3] \geq (\kappa/2)^{N_{\mathcal{P}}} = (\Theta_{\mathcal{P}}(1))^m$.

These four events concern only arms 1 and 2; since each arm $a \in \{3, \ldots, K\}$ has an independent prior and reward tape, the events $\texttt{Ev}_a = \{\mu_a \in (\tau, 1 - \tau)\}$ are independent of events (i)–(iv) above and of each other. Multiplying all probabilities:

$$\Pr[\texttt{FAIL}_{c_{\mathcal{P}}, N_{\mathcal{P}}}] \geq \frac{\Delta}{8} \cdot \Theta_{\mathcal{P}}(1) \cdot (\Theta_{\mathcal{P}}(1))^m \cdot \prod_{a=3}^{K} \Pr[\texttt{Ev}_a^+] = \Theta_{\mathcal{P}}(1)^m > 0.$$

Absorbing all prior-dependent constants into $q_{\mathcal{P}}$ gives $\Pr[\texttt{FAIL}_{c_{\mathcal{P}}, N_{\mathcal{P}}}] \geq q_{\mathcal{P}}^m$. $\qquad\square$

## H.2. Proof of Corollary H.3

We prove Corollary H.3 by following the same failure $\to$ strong failure $\to$ regret chain as in Section 3.

**Definition H.7.** Fix $c > 0$, $N \in \mathbb{N}$, $K \geq 3$. Event $\texttt{StrongFail}_{c,N}$ occurs if:

- *(Arm 2 dominates)* $\mu_2 \geq \mu_a + c$ for every $a \neq 2$.

- *(Arm 2 rarely considered)* At most $N$ episodes consider arm 2.

**Lemma H.8** (K-arm weak-to-strong)**.** *Fix $c \in (0, 1/2)$ and $N \in \mathbb{N}$. Let $\delta := \Pr[\texttt{FAIL}_{c,N}] > 0$. Then*

$$\Pr\left[\texttt{StrongFail}_{c,N'} \mid \texttt{FAIL}_{c,N}\right] \geq \tfrac{1}{2},$$

*for some $N' < \infty$ determined by $c$, $N$, $\delta$, and $m$.*

*Proof.* Similar to the proof of Lemma 3.4. Let $\mathcal{E} := \{\mu_a \in (c, 1 - c) \text{ for all } a \in [K]\}$; since $\mathcal{E}$ is condition (i) of $\texttt{FAIL}_{c,N}$, it holds whenever $\texttt{FAIL}_{c,N}$ does, so conditioning on $\mathcal{E}$ does not alter the conditional probability. In any good episode (one that considers arm 2), arm 2 is pulled with probability at least $c^m$, using $\mu_a \in (c, 1 - c)$ for all $a$ exactly as in the original proof. The concentration argument (Lemma A.1) and the choice of $N'$ are unchanged. $\qquad\square$

**Lemma H.9** (K-arm utility gap)**.** *Fix $c > 0$. Consider a realized instance with $\mu_2 \geq \mu_a + c$ for every $a \neq 2$ and $\mu_a \in (c, 1 - c)$ for all $a \in [K]$. Then*

$$c_{\text{sel}} \leq \frac{c'}{2m} \implies G(\vec{\mu}) \geq \frac{c'}{2},$$

*where $c' := c^{2m}(f(\vec{1}) - f(\vec{0})) > 0$ depends only on $c$ and $f$.*

*Proof.* Let $a^* := \arg\max_{a \neq 2} \mu_a$; then $\mu_2 \geq \mu_{a^*} + c$. The proof of Lemma 3.8 applies with arm 1 replaced by arm $a^*$ throughout. The coupling uses $\mu_2 \geq \mu_{a^*} + c$ and proceeds similarly: with probability $\geq c^{2m}$ we have $\vec{r}^{(2)} = \vec{1}$ and $\vec{r}^{(a^*)} = \vec{0}$ simultaneously, giving $V'(\pi_2') - V'(\pi_{a^*}') \geq c'$. The bound $V'(\pi_{a^*}') \geq V'(\pi)$ for all $\pi \in \Pi_{\text{bad}}$ holds because skip is dominated by any arm with positive mean reward, and arm $a^*$ has the highest mean among all arms $a \neq 2$, so $\pi_{a^*}'$ maximizes $V'$ over $\Pi_{\text{bad}}$. The cost adjustment is similar: $c_{\text{sel}} \leq c'/(2m)$ implies $G(\vec{\mu}) \geq c' - m c_{\text{sel}} \geq c'/2$. $\qquad\square$

*Proof of Corollary H.3.* We apply the chain from Section 3 consecutively. Theorem H.2 gives $\Pr\left[\text{FAIL}_{c_\mathcal{P}, N_\mathcal{P}}\right] \geq q_\mathcal{P}^m > 0$. Lemma H.8 gives $\Pr\left[\text{StrongFail}_{c_\mathcal{P}, N'} \mid \text{FAIL}_{c_\mathcal{P}, N_\mathcal{P}}\right] \geq \frac{1}{2}$ for some $N' < \infty$, hence $\Pr\left[\text{FAIL}_{c_\mathcal{P}, N_\mathcal{P}} \cap \text{StrongFail}_{c_\mathcal{P}, N'}\right] \geq q_\mathcal{P}^m/2$. Theorem 3.6 applies to $\text{StrongFail}_{c_\mathcal{P}, N'}$: its proof counts episodes in which arm 2 is not considered (of which there are at least $T - N'$ by definition of $\text{StrongFail}$), and in each such episode the policy is in $\Pi_{\text{bad}}$, so $\text{Reg}(T) \geq (T - N') \cdot G(\vec{\mu})$ holds deterministically. Under $\text{FAIL}_{c_\mathcal{P}, N_\mathcal{P}}$, conditions (i) and (ii) give $\mu_a \in (c_\mathcal{P}, 1 - c_\mathcal{P})$ for all $a$ and $\mu_2 \geq \mu_a + c_\mathcal{P}$ for all $a \neq 2$, so Lemma H.9 gives $G(\vec{\mu}) \geq c'/2$ for $c_{\text{sel}} \leq c'/(2m)$. Taking expectations over $\mathcal{P}$:

$$\text{BReg}(T) \;\geq\; \frac{q_\mathcal{P}^m}{2} \cdot \frac{c'}{2} \cdot (T - N').$$

Setting $c_0 := q_\mathcal{P}^m c'/4$ and $N_0 := N'$ completes the proof. $\qquad\square$

# I. Zero-Cost Model

Our analysis sheds light on the model with zero selection costs. For concreteness, we focus on $m = 2$ and a symmetric aggregation function $f$. We assume skips are allowed and ties in posterior utility are broken in favor of skips.

**Theorem I.1.** *Consider an* `EpiBSL` *instance with $m = 2$, symmetric $f$, selection cost $c_{\text{sel}} = 0$, skips allowed, and ties broken in favor of skips. Then there exist constants $N_{\mathcal{P}} < \infty$ and $c_{\mathcal{P}}, q_{\mathcal{P}} > 0$ depending only on $\mathcal{P}$ such that*

$$\Pr\left[\text{FAIL}_{c_{\mathcal{P}}, N_{\mathcal{P}}}\right] \geq q_{\mathcal{P}} > 0.$$

The following corollary derives linear Bayesian regret via the same chain as in Section 3.

**Corollary I.2.** *Under the conditions of Theorem I.1, there exist constants $c_0 > 0$ and $N_0 < \infty$ depending on $\mathcal{P}$ and $f$ such that*

$$\text{BReg}(T) \geq c_0 \cdot (T - N_0) \qquad \text{for any episode } T.$$

*This is the same form as Corollary 4.5.*

*Proof.* We apply the same chain as in the proof of Corollary 4.5: consecutively apply Lemma 3.4, Theorem 3.6, and Lemma 3.8, with Theorem I.1 in place of Theorem 4.6. Since $c_{\text{sel}} = 0$, the cost bound in Lemma 3.8 is trivially satisfied. $\square$

We prove Theorem I.1 by splitting into two cases based on whether $f = \max$ or not.

## I.1. Proof of Theorem I.1: Case $f \neq \max$

We impose the condition

$$f(1,1) > f(1,0) = f(0,1), \tag{16}$$

where the equality follows from symmetry of $f$; this excludes $f = \max$, for which $f(1,1) = f(1,0) = 1$.

We first establish the analogue of Lemma C.1 for zero cost.

**Lemma I.3** (Zero-cost no-pull). *Assume cost $c_{\text{sel}} = 0$ and condition (16). If*

$$\frac{\alpha_{2,t-1} + 1}{\alpha_{2,t-1} + \beta_{2,t-1} + 1} < \gamma_{1,t-1},$$

*then arm 2 is not selected in round $t$.*

*Proof.* We show that any deterministic policy playing arm 2 in round $t$ is strictly suboptimal. Since any randomized policy is a mixture of deterministic ones, this implies the lemma.

We first handle round $t$ being the *second* round of the episode.

**Claim I.4.** *If $t$ is the second round and $\gamma_{2,t-1} < \gamma_{1,t-1}$, then arm 2 is not selected.*

*Proof.* Since $\gamma_{1,t-1} > \gamma_{2,t-1}$ and $f(r_1, \cdot)$ is non-decreasing, playing arm 1 yields weakly higher posterior utility than arm 2. If strictly higher, the claim follows. If equal, then $f(r_1, 0) = f(r_1, 1)$, so skipping also yields the same expected score at zero cost — and ties go to skip, so arm 2 is not chosen. $\square$

For the remainder, assume $t$ is the first round. Since arm 2 is played in round $t$, its posterior does not update arm 1, so

$$\gamma_{2,t} \leq \frac{\alpha_{2,t-1} + 1}{\alpha_{2,t-1} + \beta_{2,t-1} + 1} < \gamma_{1,t-1} = \gamma_{1,t}.$$

By Claim I.4, arm 2 cannot be played in round 2. Write $\gamma_1 := \gamma_{1,t-1}$, $\gamma_2 := \gamma_{2,t-1}$, and $\gamma_1^+ := (\alpha_{1,t-1} + 1)/(\alpha_{1,t-1} + \beta_{1,t-1} + 1)$. Note that $\gamma_1^+ > \gamma_1 > \gamma_2$ (the first inequality holds because $\beta_{1,t-1} > 0$, and the second by lemma hypothesis). The posterior optimal policy plays arm 2 in round 1 and then one of four alternatives in round 2:

**Case I: always skip in round** $2$**.** Posterior utility equals $\gamma_2 f(1,0) + (1-\gamma_2)f(0,0)$. If $f(1,0) = f(0,0)$, this equals $f(0,0)$, which is the same as skipping both rounds; by tie-breaking in favor of skips, the policy would skip round 1 rather than play arm 2, a contradiction. If $f(1,0) > f(0,0)$, playing arm 1 in round 1 with skip in round 2 yields $\gamma_1 f(1,0) + (1-\gamma_1)f(0,0)$, which is strictly higher because $\gamma_1 > \gamma_2$, a contradiction.

**Case II: always play arm** $1$ **in round** $2$**.** Since $f$ is symmetric, $[arm_2, arm_1]$ has the same utility distribution as $[arm_1, arm_2]$ (the outcomes $(r_1, r_2)$ have the same joint law after swapping, by symmetry of $f$ and independence of the arms). We compare with the alternative policy $[arm_1$ in round 1, $arm_1$ if $r_1{=}1$, $arm_2$ if $r_1{=}0]$. The utilities of the two policies differ only on the event $r_1 = 1$. Conditional on $r_1 = 1$: the alternative plays arm 1 with updated posterior $\gamma_1^+$, while $[arm_1, arm_2]$ plays arm 2 with posterior $\gamma_2$. Since $\gamma_1^+ > \gamma_2$ and $f(1,1) > f(1,0)$ by (16), playing arm 1 in round 2 yields strictly higher utility. The event $r_1 = 1$ has probability $\gamma_1 > 0$, so the alternative is strictly better than $[arm_1, arm_2]$, hence strictly better than $[arm_2, arm_1]$, a contradiction.

**Case III: play arm** $1$ **in round** $2$ **if** $r_1{=}0$**; skip if** $r_1{=}1$**.** Skipping when $r_1 = 1$ is only optimal if doing so matches the best alternative, i.e., playing any arm gives no more than $f(1,0)$ in expectation. But playing arm 1 when $r_1 = 1$ yields $\gamma_1^+ f(1,1) + (1-\gamma_1^+)f(1,0) > f(1,0)$ because $\gamma_1^+ > 0$ and $f(1,1) > f(1,0)$ by (16). So skipping when $r_1 = 1$ is strictly suboptimal, a contradiction.

**Case IV: play arm** $1$ **in round** $2$ **if** $r_1{=}1$**; skip if** $r_1{=}0$**.** Skipping when $r_1 = 0$ is optimal (given ties go to skip) only if playing arm 1 gives no more than $f(0,0)$, i.e., $\gamma_1^- f(0,1) + (1-\gamma_1^-)f(0,0) \le f(0,0)$ where $\gamma_1^- := \alpha_{1,t-1}/(\alpha_{1,t-1} + \beta_{1,t-1} + 1)$. Since $f(0,1) \ge f(0,0)$, this forces $f(0,1) = f(0,0)$; by symmetry $f(1,0) = f(0,0)$ as well. The utility of $[arm_2, arm_1$ if $r_1{=}1]$ is then

$$\gamma_2\big(\gamma_1 f(1,1) + (1-\gamma_1)f(0,0)\big) + (1-\gamma_2)f(0,0) = \gamma_2\gamma_1 f(1,1) + (1-\gamma_2\gamma_1)f(0,0).$$

The alternative policy $[arm_1, arm_1$ if $r_1{=}1]$ yields

$$\gamma_1\big(\gamma_1^+ f(1,1) + (1-\gamma_1^+)f(0,0)\big) + (1-\gamma_1)f(0,0) = \gamma_1\gamma_1^+ f(1,1) + (1-\gamma_1\gamma_1^+)f(0,0).$$

Since $\gamma_1^+ > \gamma_2$ we have $\gamma_1\gamma_1^+ > \gamma_2\gamma_1$, and $f(1,1) > f(0,0)$ by (16), so the alternative is strictly better, a contradiction. $\square$

*Proof of Theorem I.1 for $f \ne \max$.* Similar to the proof of Theorem 4.6(a) in Appendix C, with Lemma C.1 replaced by Lemma I.3 throughout. $\square$

**I.2. Proof of Theorem I.1: Case $f = \max$**

We now handle $f = \max$, for which $f(1,1) = f(1,0) = f(0,1) =: x > 0$ and $f(0,0) = 0$. Condition (16) fails here, so Lemma I.3 does not apply. Instead we use a different no-pull condition based on arm 1's *pessimistic* posterior.

**Lemma I.5** (No-pull for $f = \max$). *Assume $c_{\mathrm{sel}} = 0$ and $f = \max$. Write $\gamma_1^- := \alpha_{1,t-1}/(\alpha_{1,t-1} + \beta_{1,t-1} + 1)$. If $\gamma_1^- > \gamma_{2,t-1}$, then arm 2 is not selected in round $t$.*

*Proof.* The second round follows from Claim I.4, since $\gamma_{1,t-1} > \gamma_1^- > \gamma_{2,t-1}$.

For the first round, note that for $f = \max$ the optimal policy after playing any arm $a$ in round 1 is: skip round 2 if $r_1 = 1$ (tie-break: score is already $x$ regardless), and play arm 1 in round 2 if $r_1 = 0$ (since $f(0,1) = x > 0 = f(0,0)$ and $\gamma_{1,t-1} > \gamma_{1,t-1}^- > \gamma_{2,t-1}$). Thus the only policies to compare are $[arm_2, arm_1$ if $r_1{=}0]$ and $[arm_1, arm_1$ if $r_1{=}0]$, with posterior utilities

$$U_2 = \big(1 - (1-\gamma_2)(1-\gamma_1)\big)x,$$
$$U_1 = \big(1 - (1-\gamma_1)(1-\gamma_1^-)\big)x.$$

Since $\gamma_1^- > \gamma_2$, we have $(1-\gamma_1^-) < (1-\gamma_2)$, so $(1-\gamma_1)(1-\gamma_1^-) < (1-\gamma_1)(1-\gamma_2)$, giving $U_1 > U_2$. $\square$

Recall that $\Delta = \gamma_{1,0} - \gamma_{2,0}$. Assume $N_{\mathcal{P}} > \frac{4\gamma_{2,0}}{3\Delta} - (\alpha_{1,0} + \beta_{1,0})$, and define

$$\mathtt{Ev}_2^{\max} := \{\mathtt{tape}_{2,\ell} = 0 \text{ for all } \ell \in [N_{\mathcal{P}}]\}.$$

**Lemma I.6.** *Under* $\mathrm{Ev}_1$ *and* $\mathrm{Ev}_2^{\max}$*, arm* 2 *is pulled at most* $N_{\mathcal{P}}$ *times.*

*Proof.* Suppose for contradiction that arm 2 is pulled $N_{\mathcal{P}} + 1$ times; let $t^*$ be the round of this pull. By round $t^*$, arm 2 has been pulled $N_{\mathcal{P}}$ times with all rewards 0 (by $\mathrm{Ev}_2^{\max}$), so the posterior entering round $t^*$ satisfies

$$\gamma_{2,t^*-1} = \frac{\alpha_{2,0}}{\alpha_{2,0} + \beta_{2,0} + N_{\mathcal{P}}} < \gamma_{2,0}.$$

We now characterize what happened during the $N_{\mathcal{P}}$ arm-2 pulls prior to round $t^*$, establishing a lower bound on arm-1 pulls at time $t^*$.

**Step 1: arm** 2 **only appears in first round of episodes (for all** $t \le t^*$**).** Under $\mathrm{Ev}_1$, $\gamma_{1,t} \ge \gamma_{1,0} - \Delta/4 > \gamma_{2,0}$ for all $t \le t^*$. Under $\mathrm{Ev}_2^{\max}$, arm 2's posterior only decreases with each zero-reward pull, so $\gamma_{2,t} \le \gamma_{2,0} < \gamma_{1,0} - \Delta/4 \le \gamma_{1,t}$ for all $t \le t^*$. By Claim I.4, arm 2 is never selected in round 2 of any episode up to $t^*$.

**Step 2: each arm-**2 **pull generates an arm-**1 **pull (for all** $t \le t^*$**).** In every episode prior to $t^*$ where arm 2 is pulled in round 1, it yields reward 0 (by $\mathrm{Ev}_2^{\max}$). After observing $r_1 = 0$, the score is $\max(0, r_2) = r_2$, so playing arm 1 in round 2 gives $\gamma_{1,t} \cdot x > 0$ while skipping gives 0; since $c_{\mathrm{sel}} = 0$, arm 1 is strictly preferred over skip. By Step 1, arm 2 is not played in round 2. Hence arm 1 is played in round 2 in every such episode, contributing one arm-1 pull per arm-2 pull. Over the $N_{\mathcal{P}}$ arm-2 pulls before $t^*$, arm 1 accumulates at least $N_{\mathcal{P}}$ pulls, giving $\alpha_{1,t^*-1} + \beta_{1,t^*-1} \ge \alpha_{1,0} + \beta_{1,0} + N_{\mathcal{P}}$.

**Step 3: apply Lemma I.5.** Under $\mathrm{Ev}_1$ with $\alpha_{1,t^*-1} + \beta_{1,t^*-1} \ge \alpha_{1,0} + \beta_{1,0} + N_{\mathcal{P}}$:

$$\gamma_1^- = \frac{\alpha_{1,t^*-1}}{\alpha_{1,t^*-1} + \beta_{1,t^*-1} + 1} \ge \left(\gamma_{1,0} - \tfrac{\Delta}{4}\right) \cdot \frac{\alpha_{1,0} + \beta_{1,0} + N_{\mathcal{P}}}{\alpha_{1,0} + \beta_{1,0} + N_{\mathcal{P}} + 1}.$$

Let $n := \alpha_{1,0} + \beta_{1,0} + N_{\mathcal{P}}$. The condition $N_{\mathcal{P}} > \frac{4\gamma_{2,0}}{3\Delta} - (\alpha_{1,0} + \beta_{1,0})$ is equivalent to $n \cdot \frac{3\Delta}{4} > \gamma_{2,0}$, i.e., $(\gamma_{1,0} - \tfrac{\Delta}{4} - \gamma_{2,0}) \cdot n > \gamma_{2,0}$, which in turn is equivalent to

$$\left(\gamma_{1,0} - \tfrac{\Delta}{4}\right) \cdot \frac{n}{n+1} > \gamma_{2,0}.$$

Hence $\gamma_1^- \ge \left(\gamma_{1,0} - \tfrac{\Delta}{4}\right) \cdot \frac{n}{n+1} > \gamma_{2,0} > \gamma_{2,t^*-1}$, and by Lemma I.5, arm 2 is not pulled at $t^*$, a contradiction. $\square$

*Proof of Theorem I.1 for $f = \max$.* Similar to the $f \ne \max$ case, with Lemma I.6 and Lemma I.5 replacing Lemma C.4 and Lemma I.3 respectively. $\square$

