# OpenReview forum: "Bandit Social Leaning Dynamics with Exploration Episodes"
_ICML.cc/2026/Conference — ICML 2026 regular_

### Official Review · Reviewer_jpYk · 2026-03-11

**Soundness:** 3
**Presentation:** 3
**Significance:** 3
**Originality:** 3
**Overall Recommendation:** 4
**Confidence:** 3

**Summary:**

This paper investigates a two-arm Bayesian bandit social learning model where self-interested agents arrive sequentially to make a short sequence of consecutive decisions, termed an "episode". Even though these episodic structures naturally incentivize agents to perform some within-episode exploration to maximize their own aggregate utility, the authors prove that such endogenous exploration is still insufficient for effective social learning. They demonstrate that the Bayesian regret grows linearly over time across various episode lengths and utility aggregation functions (such as the sum, max, or min of the rewards). The authors conclude that organic exploration is insufficient, and externally driven exploration remains strictly necessary to avoid suboptimal outcomes.

**Compliance With Llm Reviewing Policy:**

Affirmed.

**Final Justification:**

Thank the authors for the clear responses. My concerns have been addressed. I keep my positive score.

**Key Questions For Authors:**

- How essential is the two-arm Bernoulli/Beta setup? Do the authors expect the same typical-failure phenomenon to hold for more general reward distributions or more arms?
- For $m>2$, is the symmetry assumption mainly a proof artifact, or is it fundamentally necessary?
- Can the authors provide quantitative intuition on the failure probability and regret constants for representative priors?

**Limitations:**

yes

**Strengths And Weaknesses:**

Strengths:
- The paper is well motivated and studies a natural and important extension of prior bandit social-learning models by allowing each agent to act over an episode rather than a single round.
- The authors establish a clear and strong negative result: even with within-episode incentives to explore, social learning can still fail and incur linear Bayesian regret.
- The analysis covers a broad family of utility aggregation functions, including sum, max, and min, which strengthens the scope of the theoretical message.
- The technical development appears nontrivial and carefully structured, connecting exploration failure to stronger failure events and then to regret guarantees.

Weaknesses:
- The analysis is developed in a fairly stylized two-arm Beta-Bernoulli model with a skip arm, which makes the result clean but leaves open how broadly the conclusions extend beyond this setting.
- The most general theorem for $m>2$ still relies on symmetric utility aggregation; extending the result to richer non-symmetric utilities for longer episodes remains open.

---

> ### Author Rebuttal · Authors · 2026-03-30
>
> Thanks for the good questions. We hope our responses help clarify things (and we'll propagate these clarifications to the revision).
>
> **W1 (idealized setting)**: See our response to (Rev 6gAk, W2). In short: (i) we believe our 2-armed model suffices to resolve the high-level question that we posed, and moreover (ii) our analysis easily extends to a broad family of problem instances with >2 arms.
>
> **W2 (symmetry assumption for m>2)**: please see Q2.
>
>
> **Q1 (conjugate priors)**. Our analysis relies on explicit posterior updates, i.e., on having conjugate priors. Beta-Bernoulli conjugate priors is a very standard modeling choice throughout all Bayesian analyses in the literature.
>
> We conjecture our analysis should carry over to any bounded rewards and any priors under some mild assumptions (e.g., the posterior should be monotone in realized rewards), but we don't know how to handle the technicalities.
>
> *However*, going beyond conjugate priors stretches the motivating story, as  human agents are unlikely to engage in complex Bayesian reasoning.
>
> **Q1 (2 arms)**. Having only two non-skip arms is essential in our analysis to derive failures for _every_ bandit instance. That said, the same techniques yield failures for >2 arms, [but only] for a substantial family of problem instances, see our response to (Rev 6gAk, W2).
>
> We conjecture failures for >2 arms for "almost" all bandit instances, but proving this seems to require some new technical ideas.
>
> **Q2**: The symmetry assumption for $m>2$ is there to make the analysis go through.
>
> That said, symmetric aggregation function $f$ already allows for considerable generality: e.g., it subsumes our main motivating cases (max, min and sum) and any weighted combination thereof. We emphasize that a natural "default" choice from bandit perspective would have been $f=\sum$, and we go (far) beyond that as a robustness check. As an extension, we remove the symmetry assumption for episode length $m=2$. Doing the same for $m>2$ is a nice technical open question.
>
> **Q3**: This paper focuses on the distinction between linear vs sublinear regret as the most basic notion of success vs failure in bandit learning. In this sense, any positive-constant failure probability (FailProb), however small, implies linear regret.
>
> Our lower bounds on FailProb are only "qualitative": we only strive to guarantee "some positive constant", without attempting to optimize it.
>
> In that, we match the SoTA for 2-armed Bayesian bandits (without episodes, i.e., for m=1), from Banihashem et al. (2023). In particular, both our lower bounds and theirs decay exponentially with the "strength" of the prior (*). Also, they are proportional to the prior gap, $|E[\mu_1-\mu_2]|$, and hence they become vacuous when the prior gap goes to 0.
>
> (*) For $\text{Beta}(\alpha, \beta)$ prior, we define "strength" as $\alpha+\beta$.
>
> Improving the dependence on the prior is an important open question. However, we believe the appropriate way to address it would be for 2-armed Bayesian bandits, without episodes.

---

> > ### Author Rebuttal · Reviewer_jpYk · 2026-04-01
> >
> > Thank the authors for the clear responses. My concerns have been addressed and I maintain my positive assessment. I will consider updating my score based on the final discussion with other reviewers.

---

### Official Review · Reviewer_qVLS · 2026-03-12

**Soundness:** 3
**Presentation:** 2
**Significance:** 3
**Originality:** 3
**Overall Recommendation:** 4
**Confidence:** 2

**Summary:**

In a two-arm bandit setting, the paper shows that even when each self-interested agent receives multiple pulls within a session and can explore locally, exploration may still collapse because agents optimize their own episode utility rather than long-run social learning, so the better arm can be ignored with positive probability and social regret grows linearly.

**Compliance With Llm Reviewing Policy:**

Affirmed.

**Key Questions For Authors:**

The authors may address the weaknesses listed above, particularly those related to clarity and presentation, and comment on the other limitations. I am inclined to consider updating my score based on improvements to clarity, the authors’ responses, and the opinions of other reviewers with relevant expertise.

**Limitations:**

Yes.

**Strengths And Weaknesses:**

**Strengths**

S1. Novel extension of social bandit models.
The paper studies a setting where agents make multiple decisions within an episode, extending prior social bandit work that assumes a single decision per agent. This better captures realistic interaction patterns such as user sessions or repeated decisions.

S2. Conceptually interesting negative result.
The paper shows that within-episode exploration does not resolve exploration failures caused by self-interested agents. Even when agents can experiment locally during their session, social learning can still collapse.

S3. Rigorous theoretical analysis.
The paper provides formal proofs showing that learning failures occur with positive probability and imply linear Bayesian regret, establishing a strong theoretical limitation of decentralized exploration dynamics.




**Weaknesses:**

1. Presentation

W1. Informal terminology in the main text.
The paper introduces concepts such as “episodes consider arm 2” and “good episodes” without formal definitions in the main text. These notions are only implicitly clarified in the appendix through the construction of the variables Y_{e_i}. This creates confusion between episodes that may play an arm and those that actually play it, which is important for understanding the proofs.

W2. Proof sketches do not closely match the appendix proofs.
The proof sketches in the main text describe the argument at a high level (e.g., minimal probability of pulling arm 2 and an Azuma–Hoeffding argument), but the appendix uses a more involved construction with auxiliary random variables and padding arguments. The mapping between the sketch and the formal proof is not explicit, making it harder for readers to follow the logic.

W3. Key assumptions are only made explicit in the appendix.
Some assumptions required by the proofs appear primarily in the appendix rather than being emphasized in the main text. For example, the bounded mean condition \mu_1, \mu_2 \in (c,1-c) is critical to the probability bounds used in Lemma 3.4 but it’s meaning and implication is not highlighted clearly when the lemma is introduced.

W4. Technical constructions are not well motivated.
The auxiliary sequence Y_{e_i} used in the appendix proof is somewhat artificial and not well motivated in the main text. Without explanation, it can appear ad hoc and obscure the intuition behind the concentration argument.

⸻

2. Technical Aspects

W5. Loose constants in the analysis.
The lower bound on the probability that arm 2 is pulled in a “good” episode is derived from the probability of a specific reward sequence of length at most m, resulting in a bound of order c^m. This bound can be extremely small when m grows, leading to very loose constants and making the theoretical guarantees quantitatively weak.


⸻

3. Modeling Assumptions and Scope

W6. Two-arm setting.
The analysis largely focuses on the case of two arms. Extending the results to multiple arms could introduce additional complications and is not addressed in detail.

W7. Limited reward models.
The framework relies on Bernoulli rewards with Beta posteriors. While common in Bayesian bandit models, the proofs depend on properties of this structure, and it is not clear how directly the results extend to more general reward distributions.

W8. Fixed episode lengths.
The model assumes a fixed episode length m. In many real systems, agents may interact with the platform for varying durations or may terminate episodes early, which could affect exploration incentives.

W9. Full history observability.
Agents observe the entire past history of actions and rewards. In many practical settings, users observe only partial or aggregated information, which could significantly alter the learning dynamics.

---

> ### Author Rebuttal · Authors · 2026-03-30
>
> Thanks for raising many relevant concerns! We hope our responses help clarify things (and we'll propagate these clarifications to the revision).
>
> **WEAKNESSES: PRESENTATION**
>
> **W1**: We are not sure which terminology is informal! For the two examples you brought up, "episode considers an arm" is defined in Defn 3.2 and "good episodes" are only invoked in the proof sketch of Lemma 3.4 and defined in the first sentence thereof.
>
> The random sequence you've mentioned (the sequence of $\tilde{Y}_i$'s from L559) is not needed for any of the definitions in the body. It is just a technical gimmick inside the proof of Lemma 3.4.
>
> **W2**: Our proof sketches do not reflect the full details, as the proof sketches are supposed to do. That said, we'll flag the steps in the proof sketches which require more elaborate arguments, and we'll revise the full proofs to clarify the mapping to the sketches.
>
> **W3**: We believe that we do write out all assumptions for our theorems/lemmas in the body of the paper. The additional definitions and assumptions in the appendix only serve to prove our theorems/lemmas in the body.
>
> In particular, Lemma 3.4 contains all assumptions in its statement. The "bounded mean rewards" condition from Defn 3.1 is not an assumption in the Lemma but rather a property of the failure event being characterized. And this condition does appear in the lemma, via the FAIL event in Eq. (2). And we explain why we use this condition (right after Defn 3.1):  "for technical convenience", i.e., to simplify proofs. (This this is a quite standard condition/usage when analyzing random variables.) We'll clarify this explanation in the text, but we really hope it is clear now!
>
> **W4**: Please note that we provide a considerable amount of intuition in our proof sketches. That said, we'll revise the appendices to provide more intuition in the proofs, and we'll be happy to elaborate on any particular thing you bring up!
>
> Re the sequence of $\tilde{Y}_j$'s from L559. It is just a technical gimmick inside the proof of Lemma 3.4. Its purpose is to prove that if sufficiently may episodes consider arm $2$ then arm $2$ is chosen at least $N$ times. (We point this out in the proof sketch, and we'll recap this point in the full proof!)
>
> Let us clarify how this sequence is defined. Let $E_j$ be the $j$-th episode in which arm 2 is considered (if one exists). Then $\tilde{Y}_j$ is Indicator(episode $E_j$ exists AND arm 2 is chosen in it).
>
> **WEAKNESSES: TECHNICAL**
>
> **W5**: Our lower bound on failure probability (FailProb) decays exponentially in $m$. How the actual FailProb scales with $m$ is an open question.
>
> Our focus is small $m$, which captures realistic scenarios: e.g., try a few prompts or products.
>
> That said, we zoom in on the distinction between linear vs sublinear regret as the most basic notion of success vs failure in bandit learning. And even a tiny-but-positive FailProb is mathematically important as it implies linear regret / asymptotic failure.
>
> **WEAKNESSES: MODEL / SCOPE**
>
> **W6**, **W7**: Please see (Rev 6gAk, W2) and (Rev jpYk, Q1) re, resp., 2 arms and Beta-Bernoulli conjugate priors. In short: (i) we believe our model suffices to resolve the high-level question that we posed, and moreover (ii) our analysis easily extends to a broad family of problem instances with >2 arms.
>
> **W8**: Please note that our model allows agents to "skip" all remaining rounds (getting a reward of $0$ and no cost for each skipped round). This corresponds to terminating the episode early for paradigmatic aggregation functions such as $f=\max$ and $f=\sum$.
>
> Moreover, our analysis extends to non-uniform episode lengths upper-bounded by $m$. (With no conceptual changes but some extra messiness -- which is why we did not spell it out in the submission.)
>
> **W9**: Full observability is crucial for our results, and it is often achieved by online platforms that aggregate user reviews. Note that aggregate statistics (#pulls and average reward for each arm) suffice for our purposes.
>
> Characterizing what happens under partial observability is indeed an important open question, e.g., when each agent only receives the history data from its "friends" on a social network. However, it is really not understood even for the basic case of 2-armed bandits where each agent controls a single round. (*) So, we believe the appropriate way to pursue this question would be by focusing on this basic case.
>
> (*) The only result on this [that we are aware of] is a rather  intricate / specific construction of a social network which guarantees learning success, in (Immorlica et al., "Incentivizing Exploration with Selective Data Disclosure", EC 2020). What happens in general -- for "greedy bandit learning" on an arbitrary social network -- is very unclear.

---

> > ### Author Rebuttal · Reviewer_qVLS · 2026-04-01
> >
> > Thank you for the detailed response. As mentioned previously, I am inclined to update my score.
> > 1.	Apologies for missing the local definition of “good episode” in the proof sketch. That said, I still find the terminology somewhat confusing in practice and easy to misread. Especially the distinction between considering an arm (= the policy assigns a probability greater than zero of playing/pulling the arm at some point in the episode) and choosing or playing/pulling the arm (= the realized action). Making this distinction more explicit in the main text would improve readability and clarity.
> > 1.1	Given this, I still find the proof sketch somewhat difficult to follow without referring to the appendix. Strengthening the connection between the sketch and the formal proof would improve readability.
> > 2.	Thank you for the responses to W5–W9. I believe these points would help strengthen a more explicit discussion of the modeling assumptions and limitations in the paper.

---

### Official Review · Reviewer_6gAk · 2026-03-13

**Soundness:** 3
**Presentation:** 3
**Significance:** 2
**Originality:** 2
**Overall Recommendation:** 4
**Confidence:** 4

**Summary:**

The authors study an idealised form of a Bayesian social learning problem on bandits. Multiple players may come in sequentially to interact for a finite number of rounds with a bandit problem, based on some prior over mean rewards, and decide for different strategies in terms of the policy being used. They argue that these problems suffer from exploration failures; due to the short nature of the episodes, no agent has an incentive to explore, and all end up getting stuck in local policy minima.

At each episode, the corresponding player chooses a policy to maximize expected utility, and can either pull one of two Bernoulli arms or use a skip arm with zero reward. Using this model, the authors show that for any fixed episode length m, learning failure happens with positive probability, which implies linear Bayesian regret over time.

**Compliance With Llm Reviewing Policy:**

Affirmed.

**Final Justification:**

I maintain my positive evaluation of the paper.

**Key Questions For Authors:**

- The problem setting is not entirely clear to me yet. What is the information flow between sequential agents? Do agents inherit the prior from the previous agent's posterior? Do they inherit the history of all arm pulls in all previous episodes? Or is the history re-set per episode and the prior re-initialised to some common distribution?
- Why would these agents not follow existing no-regret algorithms? (eg Thompson Sampling). If the prior is well specified (to the corresponding true mean distribution of the underlying bandit) then a TS policy would be sampling from the actual probability of each arm being optimal, which is no-regret in the limit. Is this not an option for the agents? (perhaps I missed some key constraint in this case).
- The skip arm and explicit exploration cost are central to the model and to several proof steps. How would the negative result presented by the authors be affected if 1. skipping is disallowed or 2. if exploration costs are implicit rather than explicit.
- On this line, it's not clear to me what the role or justification for the exploration cost is. As far as I understand, this is represented in the paper by all arms (except the skip arm) having a small non-zero negative reward (cost) associated with pulling it. First, is this simply added to the rewards at every pull? Second, other classic Bayesian bandit algorithms do not consider such an explicit cost to exploration, and in some cases (eg Thompson Sampling) the distinction between exploration and exploitation is not even explicit; the agent has a prior that represents its uncertainty over the model class, samples from the prior and pulls the best corresponding arm. In this case there is no distinction between exploration and exploitation; the agent acts according to its own uncertainty. Higher uncertainty naturally induces more exploration, and the cost is implicit by having a higher entropy policy that may sample worse rewards while trying to reduce this uncertainty. Could the authors comment on how the exploration costs fit into the general Bayesian bandit framework?
- In Theorem 4.1, the failure probability is lower bounded by $q_{\mathcal{P}}^m$. Can the authors clarify how this should be interpreted for moderate or large fixed episode lengths $m$? Will $q$ decay?
- In line 136, the history is defined for $i\in[t-1]$, but $i$ is also used for the arm pulled.
- I'm slightly confused by the mean reward notation $\mu$. It seems to be used interchangeably by the empirical mean in some episode (random variable), the true expected reward by the arm (expectation over a random variable) and the mean of the rewards according to the prior.

**Limitations:**

Yes

**Strengths And Weaknesses:**

## Strengths
- The paper addresses an interesting problem in joint (social) learning settings where multiple agents may be interacting with the same environment and may have implicitly coupled utility functions through the exploration component of the problem.
- The abstracted form of the problem presented by the authors is comprehensive and helps derive intuition about social learning failure modes.
- The broader consideration of different aggregate utility functions $f$ seems like a useful framework for this context.

## Weaknesses
- Some notation or formal arguments are not clear enough in its current form (see below).
- The current abstraction to 2+1 arms and Bernoulli rewards, while enough to derive intuition, is quite restrictive and a generalisation is not obvious.
- Additional intuition is needed for interpretation of some the theoretical results.

---

> ### Author Rebuttal · Authors · 2026-03-30
>
> So many good questions, thanks! We hope our replies add clarity (and we'll clarify in the revision).
>
> Re **W2** (2 arms). Our paper answers (Q) from p1: "Can learning failures be avoided when self-interested agents make a few decisions each".
>
> Arguably, our model suffices to resolve (Q) in the negative, on a high level. First, even one counterexample rules out a general positive result. What we show is much stronger: failure is ubiquitous for the paradigmatic setting of two arms (indeed, it happens for any bandit instance, any episode length, and any symmetric aggregation function). Therefore, "learning success", if any, must be confined to some exotic regimes which completely exclude this paradigmatic setting.
>
> NOTE: idealized models (e.g., 2 arms) are very common in social learning (no room to add citations, unfortunately).
>
> That said, our analysis easily extends to a broad family of problem instances with >2 arms where two arms look much better than all others according to the prior. (The sufficient condition depends only on the prior and $m$.) Our analysis deals with some events for these two arms that cause a failure, and under these events [our analysis immediately implies that] all other arms are never chosen. This extension felt redundant for answering (Q), but we can include it in a revision.
>
> **Q1**: Each agent observes the history of all previous agents (L135-137; we'll clarify). With a common prior and no private signals, this is equivalent to starting from the last agents' posterior.
>
> **Q2**: Each agent acts in its own interest, entirely ignoring the other agents' interests. This paper, along with many others in EconCS, studies the consequences of such behavior.
>
> So, while Thompson Sampling may be reasonable for a hypothetical "social planner", each agent maximizes hew own expected utility across her episode, and chooses some "per-episode policy" to achieve that.
>
> No-regret algorithms such as TS are only _asymptotically_ optimal, and usually not that good at small #rounds. Running TS within a particular episode is feasible but (very) suboptimal unless episode length $m$ is huge. In particular, without episodes (i.e., $m=1$), the optimal policy simply chooses the best arm according to the posterior.
>
> **Q3**: We believe the model with costs and skips reflects the motivating applications (see Q4), and should be our main model.
>
> That said, our analysis sheds light on the model with zero costs. For concreteness, focus on $m=2$ and symmetric aggregation function $f$. Suppose skips are allowed and ties are broken in favor of skips. Then our analysis carries over as is whenever $f\neq \max$. For $f=\max$, the results stands, but with some changes to the proofs.
>
> Without skips, the costs don't matter (each agent incurs the same cost per round no matter what) and our failure analysis seems to crumble. In fact, we have "learning success" for $f=\max$ (resp., $f=\min$) under any random tie-breaking. Indeed, if round $1$ of an episode returns reward 1 (resp., reward 0), then the subsequent round(s) does not affect agent's utility. So, we have a tie, and each arm gets chosen with some prob.
>
> We did not discuss this in the submission to keep our results clean and focused on the main model, but we can spell it out in the revision.
>
> **Q4**: The "cost" in our model represents agent's time/effort for one more pull of a non-skip arm (and yes, it is subtracted from the reward). In substance, we need costs/skips to ensure that (i) choosing an arm with a super-small reward should be strictly worse than not doing anything in this round, and so (ii) agents would not keep exploring under little/no incentive to do so. We believe this is the reality in the motivating applications.
>
> NOTE: our "cost" does not represent the FULL cost of exploration (i.e., the opportunity cost of not exploiting). So, "selection cost" may be a better term.
>
> To fit cost $x$ into a standard Bayesian bandit framework, just redefine the possible rewards of the non-skip arms as $\{-x, 1-x\}$ (while the skip arm has reward 0). We opted to separate "costs" and "rewards" for cleaner exposition.
>
> **Q5**: Our lower bound on failure probability (FailProb) decays exponentially in $m$. How the actual FailProb scales with $m$ is an open question.
>
> Interpretation:
>
> 1. our guarantee is quantitatively strong for small $m$. This regime captures realistic scenarios: e.g., try a few prompts or products.
>
> 2. Even a tiny-but-positive FailProb is mathematically important as it implies linear regret / asymptotic failure.
>
> (Cf. Rev jpYk, Q3)
>
> **Q6**: Apologies! In L136, the history should be
>     $H_t = ( (i_s, r_s): s\in[t-1] )$,
> a standard notion in bandits.
>
> **Q7**: $\mu_i$ is the mean reward of arm $i$ when this arm is pulled. Initially, it is drawn from a prior (see L152--155). We write $E[\mu_i]$ to denote the prior mean reward (e.g., L174). Where did we use $\mu_i$ inconsistently? Pls provide specific line #s, we'll fix!

---

> > ### Author Rebuttal · Reviewer_6gAk · 2026-04-02
> >
> > I thank the authors for the detailed answers. My remaining doubts are a matter of subjective interpretation of validity and relevance of assumptions, eg Q3-Q4. I am overall positive and will consider raising my score if appropriate after internal discussions.

---

### Decision · Program_Chairs · 2026-04-30

**Decision:**

Accept (regular)

**Comment:**

This paper studies a Bayesian bandit social learning model where self-interested agents arrive sequentially to make a short "episode" of consecutive decisions. Though such episodic structures naturally incentivize agents to perform intra-episode exploration to maximize their own aggregate utility, this paper shows that such endogenous exploration is still insufficient for effective social learning, and externally driven exploration remains strictly necessary to avoid suboptimal outcomes.

As the reviewers have pointed out, this paper is a novel extension of social bandit models by studying a setting where agents make multiple decisions within an episode. It presents conceptually very interesting results and provides rigorous theoretical analysis. All reviewers recommend accepting this paper. Moreover, all reviewers acknowledge that their concerns have been fully resolved after the authors' rebuttals.

After reading the paper, the reviews, and the rebuttals, I agree with the reviewers and recommend accepting this paper.